# Rate-Sensing Performance of Imperfect Capacitive Ring-Based MEMS Coriolis Vibrating Gyroscopes at Large Drive Amplitudes [note 1]

**DOI:** 10.3390/s25072263

**Published:** 2025-04-03

**Authors:** Davin Arifin, Stewart McWilliam

**Affiliations:** Department of Mechanical, Materials and Manufacturing Engineering, University of Nottingham, University Park, Nottingham NG7 2RD, UK

**Keywords:** Coriolis vibrating gyroscope, electrostatic nonlinearities, electrostatic tuning, force balancing

## Abstract

This paper investigates the effect of electrostatic nonlinearity on the rate-sensing performance of imperfect ring-based Coriolis Vibrating Gyroscopes (CVGs) for devices having 8 and 16 evenly distributed electrodes. Mathematical models are developed for CVGs operating in (i) an open loop for a linear electrostatically trimmed device, (ii) a closed loop where a sense force balancing is applied to negate the sense quadrature response, and the effects of electrostatic nonlinearity are investigated for increasing drive amplitudes. The modeling indicates the nonlinear responses for 8- and 16-electrode arrangements are quite different, and this can be attributed to the nonlinear frequency imbalance, which depends on the drive and sense frequency softening as well as the presence of self-induced parametric excitation in the sense response. In open loop the 16-electrode arrangement exhibits much weaker levels of nonlinearity than the 8-electrode arrangement because the nonlinear frequency imbalance is less sensitive to drive amplitude. For devices operating in closed-loop with sense force balancing to ensure the drive and sense responses are in-phase/anti-phase, it is shown that ideal rate-sensing performance is achieved at large drive amplitudes for both 8- and 16-electrode arrangements. Using sense force balancing, rate sensing can be achieved using either the sense response or the required balancing voltage. For the latter, large nonlinear frequency imbalances and low damping levels enhance rate-sensing performance.

## 1. Introduction

The operation of MEMS Coriolis Vibrating Gyroscopes (CVGs) fundamentally relies on the coupling of two vibrational modes due to energy transfer in the presence of an angular rate. Rings [1,2,3] are often used as the core vibrating element of these devices due to the symmetric pairing of the vibrational modes, and the planar structure also results in good compatibility with conventional MEMS fabrication techniques [4]. The in-plane flexural modes of rings occur as degenerate pairs, and this degeneracy is exploited to maximize the energy transfer between the modes via resonant operation. In practice, the primary mode, also known as the drive mode, is forced into vibration at resonance [5], and its amplitude and phase are fixed and controlled using automatic gain control (AGC) and phase-locked loop (PLL) [5,6,7]. When the device is operated in an open loop, the response of the secondary mode (also known as the sense mode) is used as a measure of the applied angular rate. Standard CVG operation involves the following sense response characteristics [5,7]: (i) the sense amplitude scales proportionally with drive amplitude; (ii) the sense amplitude scales proportionally with angular rate; (iii) the sense mode responds in phase/antiphase with the drive mode. When a device is operated in a closed loop, direct balancing forces are applied to the sense mode to counteract the Coriolis force and nullify the sense response. This approach is referred to as force-to-rebalance (FTR) [1,7], and the balancing force serves as a measure of the angular rate instead of the sense response.

In practice, ring imperfections such as material defects and manufacturing tolerances can degrade device performance. The ring imperfections result in inhomogeneous mass and stiffness distributions, which interfere with the ring dynamics, diminish the rate sensitivity, and induce quadrature and bias rate errors [7] whereby the sense response does not nullify in the absence of angular rate. Several trimming methods have been developed to mitigate these effects. In [8,9,10], electrostatic tuning is implemented by applying tuning voltages to unequally modify the stiffnesses of the drive and sense modes, aiming to achieve a net frequency split of zero. However, the linear elastic coupling due to imperfections cannot be eliminated when only eight electrodes are used [9]. In [11], the addition or removal of trimming masses is considered to address the frequency split, with detailed investigations into the requirements regarding the number and locations of the trimming masses on the imperfect ring. The common basis of these approaches involves introducing artificial imperfections to negate pre-existing ring imperfections aimed at correcting the mass and/or linear stiffness matrices of the drive and sense modes. These approaches typically focus on devices operating in the linear regime, where the modes are linearly coupled, but in practice, nonlinearities are often present. Research on the effects of electrostatic nonlinearity on tuning procedures is not widespread. One investigation into rate-integrating gyroscopes shows that quadrature nulling can be strongly affected by nonlinear behavior as the vibration amplitude increases [12], and based on the mathematical model, there is reason to believe that such corrective procedures can also be prone to nonlinearities in rate-measuring CVGs.

MEMS capacitive ring CVGs are particularly susceptible to electrostatic nonlinearities caused by narrow capacitive gaps [13]. These effects become increasingly significant as ring displacement increases relative to the gap, and the resulting effects are amplitude-dependent resonant frequencies and self-induced parametric excitation [13,14,15]. In [13], the effects of electrostatic nonlinearities on potential amplification of the sense response due to self-induced parametric excitation are investigated for the perfect ring case operating in open loop. In [14], the effects of ring imperfections in conjunction with electrostatic nonlinearity are explored, revealing that self-induced parametric excitation can enhance the sense response and rate sensitivity, but not without introducing bias rate and quadrature errors. This undermines the advantages of Coriolis force amplification at large drive amplitudes, necessitating compensation to improve rate-sensing performance. This forms the motivation for the current work, which ultimately aims to achieve ideal rate-sensing performance at increased drive amplitudes and enable Coriolis force amplification.

The work focuses on imperfect MEMS capacitive ring-based CVGs having 8- and 16-electrode arrangements where the drive amplitude is sufficiently large to exhibit electrostatic nonlinearity. Mathematical models are developed to investigate devices operating in open loop and closed loop. In the open loop, the device is linearly electrostatically trimmed, whilst in the closed loop, sense force balancing is applied to negate the quadrature sense force imbalance. This is not the same as FTR because the sense response is not nullified. The advantage of the closed loop approach used, as will be shown, is that ideal rate-sensing performance can be recaptured at large drive amplitude even when the 8-electrode arrangement is used. This is advantageous from a manufacturing standpoint because it is easier to manufacture an 8-electrode arrangement than a 16-electrode arrangement. The potential to use the applied sense force balancing instead of the sense response to measure angular rate is also demonstrated, along with identifying the necessary conditions to enhance rate-sensing performance. This article is a revised and expanded version of a paper entitled ‘Nonlinear Performance Enhancement of Imperfect Ring-based Coriolis Vibratory Gyroscopes’, which was presented at the International Conference on Recent Advances in Structural Dynamics in Southampton, UK, in July 2024 [16]. This article expands on [16] this by focusing on comparisons between 8- and 16-electrodes for both linear electrostatic tuning and sense force balancing. Additionally, this article introduces the concept of using sense force balancing for rate measurement.

The paper is organized as follows. In Section 2, the basic layout of a ring-based CVG is described, and the electrostatic configuration used to operate the device is introduced. In Section 3, the nonlinear governing equations of motion for the drive and sense responses are developed for the 8- and 16-electrode arrangements. These equations have a similar form to those in [14], but include additional electrostatic force contributions for linear electrostatic tuning and sense force balancing. These equations are used in later sections to investigate linear electrostatic tuning and sense force balancing approaches separately. Section 4 investigates the equations of motion for the linear electrostatic tuning case (with sense force balancing neglected) and the resulting open loop rate-sensing performance is assessed in terms of rate sensitivity, bias rate, and quadrature error. Section 5 investigates the equations of motion for the closed loop sense force balancing scheme (with linear electrostatic tuning neglected) in a similar way to Section 4. The rate-sensing performance is also investigated to demonstrate how the applied sense force can be used to measure angular rate. For both Section 4 and Section 5, numerical results are obtained using the Finite Element (FE) method to validate the theoretical findings. Section 6 summarizes the conclusions of the work.

## 2. Device Description

Figure 1 and Figure 2a,b show schematic diagrams of the ring-based devices considered. The ring is supported by eight uniformly-spaced supports, and surrounded by 8 and 16 evenly-distributed inner and outer electrodes. These devices are typical of those used in applications [1,9] and the operation of the 2θ in-plane flexural modes are used as the drive and sense modes, as shown. The dimensions for the ring mean radius R, radial thickness h, capacitive gap $g_{0}$, and electrode angular span δ are also shown. In practice, the capacitive gap is small ($g_{0}\ll R$) and the ring is thin (h≪R).

X and Y are generalized coordinates describing the drive and sense responses, respectively. The radial and tangential displacements of the ring centerline, u and v, at location θ, can be described using the generalized coordinates, given by the following:(1a)u=Xcos⁡2θ+Ysin⁡2θ(1b)$$v=-\frac{X}{2}\sin2\theta +\frac{Y}{2}\cos2\theta$$

This two-mode representation of the radial and tangential displacements has been widely adopted in the analysis of these device [1], and is valid for the case where the ring is thin and the resonance of the higher-order flexural modes occurs at frequencies significantly higher than the 2θ resonant frequency.

In practice, a voltage is applied to each electrode for the purposes of driving and sensing the ring displacements and tuning the ring modes [1,9]. To model the voltages applied to each electrode, each outer/inner electrode is identified by an integer index number i. The corresponding mean angular position for the ith outer/inner electrode is $\theta =\theta_{0}\left(i\right)=2i\pi /j$, where i=1,⋯,j, and j=8, 16 is the total number of electrodes within the inner or outer electrode sets. The electrode configuration is shown in Figure 2a,b.

The voltages applied in this study include components to enable linear electrostatic tuning and sense force balancing. At this stage, both voltage sources are included in the analysis. However, in later sections, the linear electrostatic tuning and sense force balancing will be implemented separately, achieved by setting relevant voltage components to zero. With the ring grounded, the voltage applied to the *i*’th inner/outer electrode having the mean electrode angular position $\theta_{0}\left(i\right)=2i\pi /j$ is as follows:(2a)$$V^{+}\left(i\right)=V_{0}+a_{AC}^{+}V_{AC}\cos\omega t\cos2\theta_{0}\left(i\right)+a_{4c}^{+}V_{4c}\cos4\theta_{0}\left(i\right)+a_{4s}^{+}V_{4s}\sin4\theta_{0}\left(i\right)\;+V_{b}\cos\frac{\omega t+\phi_{x}+i_{b}\pi }{2}\cos\left[\theta_{0}\left(i\right)-\frac{\pi }{4}\right]$$(2b)$$V^{-}\left(i\right)=V_{0}+a_{AC}^{-}V_{AC}\cos\omega t\cos2\theta_{0}\left(i\right)+a_{4c}^{-}V_{4c}\cos4\theta_{0}\left(i\right)+a_{4s}^{-}V_{4s}\sin4\theta_{0}\left(i\right)\;+V_{b}\sin\frac{\omega t+\phi_{x}+i_{b}\pi }{2}\cos\left[\theta_{0}\left(i\right)-\frac{\pi }{4}\right]$$
where ‘+’ and ‘−’ superscripts denote the outer and inner electrode sets, respectively. In these equations, $V_{0}$ is the constant preselected bias voltage component applied to all electrodes. $a_{AC}^{\pm }V_{AC}$ define the drive voltage amplitudes for the outer and inner electrodes used to drive the ring to vibrate in its 2θ mode at frequency ω. $a_{AC}^{\pm }$ are dimensionless constants defining the ratio of the drive voltage amplitudes between the inner and outer electrode and are preselected based on force conditions to be discussed later. $V_{AC}$ is varied to regulate the drive amplitude.

$a_{4c}^{\pm }V_{4c}$ and $a_{4s}^{\pm }V_{4s}$ define the linear electrostatic tuning voltage components used for linear elastic correction of the drive and sense modes. Similar to $a_{AC}^{\pm }$, $a_{4c}^{\pm }$ and $a_{4s}^{\pm }$ are dimensionless constants to be preselected based on specific force conditions to be discussed later. $V_{4c}$ and $V_{4s}$ are chosen to achieve linear electrostatic tuning.

$V_{b}$ defines the balancing voltage amplitude applied to achieve sense force balancing, and $i_{b}$ is an integer defining the balancing voltage phase index relative to the drive mode oscillation phase $\phi_{x}$. In practice, the drive voltage frequency ω is set to ensure the drive mode is resonant, such that $\phi_{x}=-\pi /2$, and the choice of $i_{b}$ is dictated by sense force balancing conditions, which will be discussed in detail later.

The following investigation uses (2a) and (2b) to conduct two separate studies: (i) linear electrostatic tuning with the balancing voltage amplitude, $V_{b},$ set to zero; and (ii) sense force balancing with $a_{4c}^{\pm }V_{4c}$ and $a_{4s}^{\pm }V_{4s}$ set to zero.

## 3. Equations of Motion

In this section, the equations of motion for the drive and sense modes of an imperfect ring using (2a) and (2b) are developed. Lagrange’s equation is used for this purpose and requires expressions for the kinetic energy and mechanical and electrostatic potential energy.

The kinetic energy of a thin ring (h/R≪1) vibrating with flexural displacement shapes shown in Figure 1 is given by the following [1]:(3)$$T=\frac{1}{2}\rho Bh\int_{0}^{2\pi }\left\{{\left(\dot{u}-v\Omega \right)}^{2}+{\left[\dot{v}+\left(R+u\right)\Omega \right]}^{2}\right\}Rd\theta \approx \frac{M}{2}\left({\dot{X}}^{2}+{\dot{Y}}^{2}\right)+\frac{4M}{5}\Omega \left(X\dot{Y}-\dot{X}Y\right)$$
where M=5/4ρπRhB is the modal mass corresponding to the 2θ modes, B is the ring axial thickness, ρ is the density of the ring material, and Ω is the angular rate to be measured. In (3), terms involving $\Omega^{2}$ have been discarded because the angular rate Ω is much smaller than the resonant frequency.

The mechanical potential energy includes contributions from ring bending [17] and the eight supports, and is given by the following:(4)$$\begin{array}{ll} U_{m}=\frac{1}{2}\int_{0}^{2\pi }E\left(\theta \right) & {\left(\frac{\partial^{2}u}{R^{2}\partial \theta^{2}}+\frac{u}{R^{2}}\right)}^{2}Rd\theta +\sum_{i=1}^{8}\left[\frac{k_{uu}{u\left(\theta_{k,i}\right)}^{2}}{2}+k_{uv}u\left(\theta_{k,i}\right)v\left(\theta_{k,i}\right)+\frac{k_{vv}{v\left(\theta_{k,i}\right)}^{2}}{2}\right]\\ & =\left[\frac{9E_{0}I\pi }{2R^{3}}\left(1+\frac{\delta E\cos4\Theta_{m}}{2}\right)+\frac{4k_{uu}+k_{vv}}{2}\right]X^{2}+\frac{9E_{0}I\pi }{2R^{3}}\delta E\sin4\Theta_{m}XY\\ & +\left[\frac{9E_{0}I\pi }{2R^{3}}\left(1-\frac{\delta E\cos4\Theta_{m}}{2}\right)+\frac{4k_{uu}+k_{vv}}{2}\right]Y^{2} \end{array}$$
where $E\left(\theta \right)$ is the inhomogeneous elastic modulus of the ring material. $E\left(\theta \right)$ can be expressed as a Fourier series in θ, where $E_{0}$ is the axisymmetric component and δE is proportional to the magnitude of the 4θ variation component in $E\left(\theta \right)$. $\Theta_{m}$ is the orientation of the 4θ variation relative to the closest electrode and represents the drive misalignment. δE and $\Theta_{m}$ are the elastic imperfection parameters breaking the symmetry of the mechanical potential energy. The summation term in (4) represents contributions from the support structures. $k_{uu}$, $k_{uv}$, and $k_{vv}$ are linear stiffness coefficients of the supports which depend on the geometry of the support. $\theta_{k,i}$ is the angular position of the ith support. The use of eight evenly spaced supports maintains the elastic symmetry [18] and contributes to ring bending to increase the net stiffness of the system.

The electrostatic potential energy is obtained using a parallel plate formulation [13,17,19] due to the narrow capacitive gaps ($g_{0}/R\ll 1$), such that the ring curvature effects are negligible. The electrostatic potential energy is given by the following:(5)$$\begin{array}{ll} U_{V}=-\frac{1}{2}\sum_{i=1}^{j} & \left({V^{+}\left(i\right)}^{2}\int_{\theta_{0}\left(i\right)-\frac{\delta }{2}}^{\theta_{0}\left(i\right)+\frac{\delta }{2}}\frac{\varepsilon_{0}BR}{g_{0}-u}d\theta \right)-\frac{1}{2}\sum_{i=1}^{j}\left({V^{-}\left(i\right)}^{2}\int_{\theta_{0}\left(i\right)-\frac{\delta }{2}}^{\theta_{0}\left(i\right)+\frac{\delta }{2}}\frac{\varepsilon_{0}BR}{g_{0}+u}d\theta \right)\\ & \;=-\frac{\varepsilon_{0}BR}{2g_{0}}\sum_{i=1}^{j}\left\{{V^{+}\left(i\right)}^{2}\sum_{m=0}^{4}\left[\int_{\frac{2i\pi }{j}-\frac{\delta }{2}}^{\frac{2i\pi }{j}+\frac{\delta }{2}}{\left(\frac{u}{g_{0}}\right)}^{m}d\theta \right]+{V^{-}\left(i\right)}^{2}\sum_{m=0}^{4}\left[\int_{\frac{2i\pi }{j}-\frac{\delta }{2}}^{\frac{2i\pi }{j}+\frac{\delta }{2}}{\left(-\frac{u}{g_{0}}\right)}^{m}d\theta \right]\right\} \end{array}$$
where $\varepsilon_{0}$ is the permittivity of vacuum. $\varepsilon_{0}BR/\left(g_{0}\pm u\right)$ is the capacitance of the inner/outer electrodes at an arbitrary angular position. In (5), a 4th order Taylor series expansion in the radial displacement u has been used to account for nonlinear electrostatic forces up to cubic order, similar to previous analysis [13,14].

Energy expressions (3)–(5) are used in Lagrange’s equation given by the following:(6)$$\frac{d}{dt}\left(\frac{\partial T}{\partial \dot{q}}\right)-\frac{\partial T}{\partial q}+\frac{\partial V}{\partial \dot{q}}+\frac{\partial U}{\partial q}=0$$
where generalized coordinate q=X or Y. In (6), V is the energy dissipation rate accounting for damping mechanisms. Evaluating (6) for q=X and Y, the governing equations of motion for the drive and sense modes can be expressed as follows:(7a)$$\begin{array}{ll} \ddot{X}+2\Gamma \dot{X} & +\left[{\omega_{m}}^{2}\left(1+\Delta_{m}\cos4\Theta_{m}\right)+\left(a_{4c}^{+}+a_{4c}^{-}\right)\alpha_{0,4c}-{\omega_{0,4c,4s,b}}^{2}+\kappa_{0,4c,4s,b}\frac{Y^{2}}{{g_{0}}^{2}}\right]X\\ & \;\;\;+\left[{\omega_{m}}^{2}\Delta_{m}\sin4\Theta_{m}+\left(a_{4s}^{+}+a_{4s}^{-}\right)\left(\alpha_{0,4s}+\kappa_{0,4s}\frac{X^{2}}{{g_{0}}^{2}}\right)\right]Y+\left[\gamma_{0,4c,4s,b}+\left(a_{4c}^{+}+a_{4c}^{-}\right)\gamma_{0,4c}\right]\frac{X^{3}}{{g_{0}}^{2}}\\ & \;\;\;+\left(a_{4s}^{+}+a_{4s}^{-}\right)\gamma_{0,4s}\frac{Y^{3}}{{g_{0}}^{2}}\\ & \;\;\;=G_{\Omega }\Omega \dot{Y}+\left[\left(a_{AC}^{+}-a_{AC}^{-}\right)\chi_{0,AC}+\left(a_{AC}^{+}a_{4c}^{+}-a_{AC}^{-}a_{4c}^{-}\right)\chi_{4c,AC}\right]\cos\omega t\left(1+c_{X^{3}}\frac{X^{2}}{{g_{0}}^{2}}+c_{XY^{2}}\frac{Y^{2}}{{g_{0}}^{2}}\right)\\ & \;\;\;+2\left[\chi_{b}\cos\left(\omega t+\phi_{x}+i_{b}\pi \right)+\left(a_{AC}^{+}a_{4s}^{+}-a_{AC}^{-}a_{4s}^{-}\right)\chi_{4s,AC}\cos\omega t\right]c_{X^{2}Y}\frac{XY}{{g_{0}}^{2}} \end{array}$$(7b)$$\begin{array}{ll} \ddot{Y}+2\Gamma \dot{Y} & +\left[{\omega_{m}}^{2}\left(1-\Delta_{m}\cos4\Theta_{m}\right)-\left(a_{4c}^{+}+a_{4c}^{-}\right)\alpha_{0,4c}-{\omega_{0,4c,4s,b}}^{2}+\kappa_{0,4c,4s,b}\frac{X^{2}}{{g_{0}}^{2}}\right]Y\\ & \;\;\;+\left[{\omega_{m}}^{2}\Delta_{m}\sin4\Theta_{m}+\left(a_{4s}^{+}+a_{4s}^{-}\right)\left(\alpha_{0,4s}+\kappa_{0,4s}\frac{Y^{2}}{{g_{0}}^{2}}\right)\right]X+\left[\gamma_{0,4c,4s,b}-\left(a_{4c}^{+}+a_{4c}^{-}\right)\gamma_{0,4c}\right]\frac{Y^{3}}{{g_{0}}^{2}}\\ & \;\;\;+\left(a_{4s}^{+}+a_{4s}^{-}\right)\gamma_{0,4s}\frac{X^{3}}{{g_{0}}^{2}}\\ & \;\;\;=-G_{\Omega }\Omega \dot{X}+2\left[\left(a_{AC}^{+}-a_{AC}^{-}\right)\chi_{0,AC}+\left(a_{AC}^{+}a_{4c}^{+}-a_{AC}^{-}a_{4c}^{-}\right)\chi_{4c,AC}\right]\cos\omega t\left(c_{XY^{2}}\frac{XY}{{g_{0}}^{2}}\right)\\ & \;\;\;+\left[\chi_{b}\cos\left(\omega t+\phi_{x}+i_{b}\pi \right)+\left(a_{AC}^{+}a_{4s}^{+}-a_{AC}^{-}a_{4s}^{-}\right)\chi_{4s,AC}\cos\omega t\right]\left(1+c_{X^{2}Y}\frac{X^{2}}{{g_{0}}^{2}}+c_{Y^{3}}\frac{Y^{2}}{{g_{0}}^{2}}\right) \end{array}$$

These equations have been normalized with respect to the modal mass M and terms involving ${V_{AC}}^{2}$ have been neglected because the drive voltage amplitude is assumed to be small. Γ is the linear damping coefficient associated with the energy dissipation rate of the ring. $\omega_{m}$ is the mechanical component of the linear resonant frequency associated with the bending potential energy of the ring and supports. $\Delta_{m}$ defines the magnitude of the mechanical imperfection. $G_{\Omega }=8/5$ is the gyroscopic coupling constant. The remaining terms in (7a) and (7b) are associated with electrostatic forces, where subscripts ‘0’, ‘AC’, ‘b’, ‘4c’, and ‘4s’ are used to identify the contributing voltage components in (2a) and (2b). Coefficients ${\omega_{0,4c,4s,b}}^{2}$, $\alpha_{0,4c}$, and $\alpha_{0,4s}$ are given by the following:(8)$$\begin{array}{l} {\omega_{0,4c,4s,b}}^{2}\\ =\left\{\begin{array}{cc} \frac{16\varepsilon_{0}\delta }{5\rho h\pi {g_{0}}^{3}}\left[2{V_{0}}^{2}+\frac{{V_{b}}^{2}}{2}+\left({a_{4c}^{+}}^{2}+{a_{4c}^{-}}^{2}\right){V_{4c}}^{2}\right], & j=8\\ \frac{32\varepsilon_{0}\delta }{5\rho h\pi {g_{0}}^{3}}\left[2{V_{0}}^{2}+\frac{{V_{b}}^{2}}{2}+\left({a_{4c}^{+}}^{2}+{a_{4c}^{-}}^{2}\right)\frac{{V_{4c}}^{2}}{2}+\left({a_{4s}^{+}}^{2}+{a_{4s}^{-}}^{2}\right)\frac{{V_{4s}}^{2}}{2}\right], & j=16 \end{array}\right. \end{array}$$(9)$$\alpha_{0,4c}=-\frac{16\varepsilon_{0}\sin2\delta }{5\rho h\pi {g_{0}}^{3}}V_{0}V_{4c}$$(10)$$\alpha_{0,4s}=\left\{\begin{array}{cc} 0, & j=8\\ -\frac{16\varepsilon_{0}\sin2\delta }{5\rho h\pi {g_{0}}^{3}}V_{0}V_{4s}, & j=16 \end{array}\right.$$

$\omega_{0,4c,4s,b}$ contributes to the linear resonant frequency, resulting in a modal-symmmetric electrostatic softening. $\alpha_{0,4c}$ represents the modal-asymmetric electrostatic forces. $\alpha_{0,4s}$ is the linear electrostatic coupling. In (10), the linear electrostatic coupling is zero for the 8-electrode arrangement because $\sin4\theta_{0}\left(i\right)=\sin8i\pi /j=0$ for all i when j=8.

Coefficients $\gamma_{0,4c,4s,b}$, $\gamma_{0,4c},$ and $\gamma_{0,4s}$ dictate the cubic electrostatic forces associated with $X^{3}$ and $Y^{3}$, given by the following:(11)$$\begin{array}{l} \gamma_{0,4c,4s,b}\\ =\left\{\begin{array}{cc} -\frac{8\varepsilon_{0}}{5\rho h\pi {g_{0}}^{3}}\left(3\delta +\frac{\sin4\delta }{4}\right)\left[2{V_{0}}^{2}+\frac{{V_{b}}^{2}}{2}+\left({a_{4c}^{+}}^{2}+{a_{4c}^{-}}^{2}\right){V_{4c}}^{2}\right], & j=8\\ -\frac{16\varepsilon_{0}}{5\rho h\pi {g_{0}}^{3}}\left[\begin{array}{c} 3\delta \left(2{V_{0}}^{2}+\frac{{V_{b}}^{2}}{2}\right)+\left(3\delta +\frac{\sin4\delta }{4}\right)\left({a_{4c}^{+}}^{2}+{a_{4c}^{-}}^{2}\right)\frac{{V_{4c}}^{2}}{2}\\ +\left(3\delta -\frac{\sin4\delta }{4}\right)\left({a_{4s}^{+}}^{2}+{a_{4s}^{-}}^{2}\right)\frac{{V_{4s}}^{2}}{2} \end{array}\right], & j=16 \end{array}\right. \end{array}$$(12)$$\gamma_{0,4c}=-\frac{32\varepsilon_{0}\sin2\delta }{5\rho h\pi {g_{0}}^{3}}V_{0}V_{4c}$$(13)$$\gamma_{0,4s}=\left\{\begin{array}{cc} 0,\; & j=8\\ -\frac{16\varepsilon_{0}\sin2\delta }{5\rho h\pi {g_{0}}^{3}}V_{0}V_{4s}, & \;j=16 \end{array}\right.$$

Coefficients $\kappa_{0,4c,4s,b}$ and $\kappa_{0,4s}$ dictate the cubic electrostatic forces associated with non-linear mode coupling $X^{2}Y$ and $XY^{2}$, given by the following:(14)$$\begin{array}{l} \kappa_{0,4c,4s,b}\\ =\left\{\begin{array}{cc} -\frac{8\varepsilon_{0}}{5\rho h\pi {g_{0}}^{3}}\left(3\delta -\frac{3\sin4\delta }{4}\right)\left[2{V_{0}}^{2}+\frac{{V_{b}}^{2}}{2}+\left({a_{4c}^{+}}^{2}+{a_{4c}^{-}}^{2}\right){V_{4c}}^{2}\right], & \;j=8\\ -\frac{16\varepsilon_{0}}{5\rho h\pi {g_{0}}^{3}}\left[\begin{array}{c} 3\delta \left(2{V_{0}}^{2}+\frac{{V_{b}}^{2}}{2}\right)+\left(3\delta -\frac{3\sin4\delta }{4}\right)\left({a_{4c}^{+}}^{2}+{a_{4c}^{-}}^{2}\right)\frac{{V_{4c}}^{2}}{2}\\ +\left(3\delta +\frac{3\sin4\delta }{4}\right)\left({a_{4s}^{+}}^{2}+{a_{4s}^{-}}^{2}\right)\frac{{V_{4s}}^{2}}{2} \end{array}\right], & \;j=16 \end{array}\right. \end{array}$$(15)$$\kappa_{0,4s}=\left\{\begin{array}{cc} 0, & \;j=8\\ -\frac{48\varepsilon_{0}\sin2\delta }{5\rho h\pi {g_{0}}^{3}}V_{0}V_{4s}, & \;j=16 \end{array}\right.$$

The terms involving $\gamma_{0,4c,4s,b}$ and $\kappa_{0,4c,4s,b}$ are the dominant nonlinear electrostatic forces due to the contribution of the bias voltage. These terms contribute to nonlinear electrostatic softening and $\kappa_{0,4c,4s,b}$ contributes to self-induced parametric excitation. For the case where the tuning voltage components $V_{4c}$ and $V_{4s}$ are small, $\gamma_{0,4c,4s,b}$ and $\kappa_{0,4c,4s,b}$ are approximately equal for the 16-electrode arrangement, or if δ=π/4 for the 8-electrode arrangement (i.e., the electrode is continuous). As will be shown, this equality has significant effects on both linear electrostatic tuning and sense force balancing.

$\chi_{0,AC}$, $\chi_{4c,AC}$, $\chi_{b}$, and $\chi_{4s,AC}$ represent the amplitudes of direct harmonic forces at frequency ω and are provided in (A1)–(A4) (Appendix A). $\chi_{0,AC}$ corresponds to the primary drive force.

When selecting $a_{AC}^{+},a_{AC}^{-}$, it is important to ensure $a_{AC}^{+}-a_{AC}^{-}\neq 0$ to prevent the drive forces from the inner and outer electrodes cancelling (see (7a)). In addition, it is necessary to select $a_{AC}^{\pm }$, $a_{4c}^{\pm }$, and $a_{4s}^{\pm }$ so that $a_{AC}^{+}a_{4c}^{+}-a_{AC}^{-}a_{4c}^{-}=0$ and $a_{AC}^{+}a_{4s}^{+}-a_{AC}^{-}a_{4s}^{-}=0$ to prevent the linear electrostatic tuning voltages from interfering with the role of the drive voltage (see (7a) and (7b)). These conditions also ensure that the quadratic force correction factors $c_{X^{3}}$, $c_{X^{2}Y}$, $c_{XY^{2}}$, and $c_{Y^{3}}$ only depend on the electrode span (see (A5)–(A8)). In subsequent analysis these conditions are used in (7a) and (7b) so there is no need to consider $\chi_{4c,AC}$ and $\chi_{4s,AC}$.

## 4. Open Loop Operation with Linear Electrostatic Tuning

In this section, linear electrostatic tuning is used to restore linear elastic symmetry between the drive and sense modes similar to that reported in [8,9] and the impact of electrostatic nonlinearity on the open loop sense response is investigated as the drive amplitude is increased.

### 4.1. Conditions for Linear Electrostatic Tuning

Linear electrostatic tuning is investigated without balancing voltage and assuming the vibration amplitudes are small. In this case, all nonlinear terms are discarded from (7a) and (7b) and the equations of motion simplify to the following:(16a)$$\begin{array}{ll} {\ddot{X}}_{l}+2\Gamma {\dot{X}}_{l} & +\left[{\omega_{m}}^{2}\left(1+\Delta_{m}\cos4\Theta_{m}\right)+\left(a_{4c}^{+}+a_{4c}^{-}\right)\alpha_{0,4c}-{\omega_{0,4c,4s}}^{2}\right]X_{l}+\left[{\omega_{m}}^{2}\Delta_{m}\sin4\Theta_{m}+\left(a_{4s}^{+}+a_{4s}^{-}\right)\alpha_{0,4s}\right]Y_{l}\\ & \;\;\;=G_{\Omega }\Omega {\dot{Y}}_{l}+\left(a_{AC}^{+}-a_{AC}^{-}\right)\chi_{0,AC}\cos\omega t \end{array}$$(16b)$$\begin{array}{ll} {\ddot{Y}}_{l}+2\Gamma {\dot{Y}}_{l} & +\left[{\omega_{m}}^{2}\left(1-\Delta_{m}\cos4\Theta_{m}\right)-\left(a_{4c}^{+}+a_{4c}^{-}\right)\alpha_{0,4c}-{\omega_{0,4c,4s}}^{2}\right]Y_{l}+\left[{\omega_{m}}^{2}\Delta_{m}\sin4\Theta_{m}+\left(a_{4s}^{+}+a_{4s}^{-}\right)\alpha_{0,4s}\right]X_{l}\\ & \;\;\;=-G_{\Omega }\Omega {\dot{X}}_{l} \end{array}$$
where the ‘l’ subscript denotes the hypothetical linear forms of the drive and sense modal coordinates. With $V_{b}=0$, the notations ${\omega_{0,4c,4s,b}}^{2}$, $\kappa_{0,4c,4s,b}$, and $\gamma_{0,4c,4s,b}$ in (7a) and (7b) are replaced by ${\omega_{0,4c,4s}}^{2}$, $\kappa_{0,4c,4s}$, and $\gamma_{0,4c,4s}$, respectively, in (16a) and (16b).

Using (16a) and (16b), linear electrostatic tuning is achieved provided the following:(17)$$\left(a_{4c}^{+}+a_{4c}^{-}\right)\alpha_{0,4c}=-{\omega_{m}}^{2}\Delta_{m}\cos4\Theta_{m}$$(18)$$\left(a_{4s}^{+}+a_{4s}^{-}\right)\alpha_{0,4s}=-{\omega_{m}}^{2}\Delta_{m}\sin4\Theta_{m}$$

These are the same conditions used for electrostatic tuning in [9], where $\alpha_{0,4c}$ and $\alpha_{0,4s}$ are regarded as artifical electrostatic-induced imperfections negating ring imperfection. To satisfy (17) and (18), $V_{4c}$ and $V_{4s}$ must be appropriately adjusted. As noted previously, (18) cannot be achieved for the 8-electrode case if $\sin4\Theta_{m}\neq 0$ due to the absence of electrostatic-induced linear elastic coupling, i.e., $\alpha_{0,4s}=0$ (see (10)). In practice, the process of achieving (17) and (18) is automated, and (17) and (18) are conventionally achieved using the drive and sense frequency response plots [10], where the resonant peaks are made to coincide.

In the following, the modal dynamics of a linear electrostatic tuned device is investigated using (7a) and (7b) with (17) and (18) satisfied. The rate-sensing performance is then assessed in terms of the rate sensitivity, bias rate, and quadrature error as the drive amplitude increases.

### 4.2. Modal Dynamics

In practice, the dynamic range of the device is such that the sense amplitude is much smaller than the drive amplitude, so the drive equation of motion (7a) can be simplified by neglecting terms involving the sense response. With (17) satisfied, (7a) simplifies to the following:(19)$$\ddot{X}+2\Gamma \dot{X}+\left({\omega_{m}}^{2}-{\omega_{0,4c,4s}}^{2}\right)X+\left[\gamma_{0,4c,4s}+\left(a_{4c}^{+}+a_{4c}^{-}\right)\gamma_{0,4c}\right]\frac{X^{3}}{{g_{0}}^{2}}=\left(a_{AC}^{+}-a_{AC}^{-}\right)\chi_{0,AC}\cos\omega t\left(1+c_{X^{3}}\frac{X^{2}}{{g_{0}}^{2}}\right)$$

An approximate solution to (19) can be obtained using the method of averaging [20,21], where the averaged steady state drive response is defined as $X=x\cos\left(\omega t+\phi_{x}\right)$. Using this method it can be shown that the drive resonant frequency $\omega =\omega_{X}$ is as follows:(20)$${\omega_{X}}^{2}={\omega_{m}}^{2}-{\omega_{0,4c,4s}}^{2}+\frac{3}{4}\left[\gamma_{0,4c,4s}+\left(a_{4c}^{+}+a_{4c}^{-}\right)\gamma_{0,4c}\right]\frac{x^{2}}{{g_{0}}^{2}}$$

Using the same conditions as for the drive mode, sense equation of motion (7b) can be linearized by discarding nonlinear terms in Y due to the much smaller oscillation amplitudes. Nonlinear quadratic terms involving the drive voltage are also neglected because the drive voltage is small compared to the bias voltage. This gives the following:(21)$$\begin{array}{ll} \ddot{Y}+2\Gamma \dot{Y} & +\left({\omega_{m}}^{2}-{\omega_{0,4c,4s}}^{2}+\kappa_{0,4c,4s}\frac{X^{2}}{{g_{0}}^{2}}\right)Y\\ & \;\;\;=-G_{\Omega }\Omega \dot{X}-\left[{\omega_{m}}^{2}\Delta_{m}\sin4\Theta_{m}+\left(a_{4s}^{+}+a_{4s}^{-}\right)\alpha_{0,4s}\right]X-\left(a_{4s}^{+}+a_{4s}^{-}\right)\gamma_{0,4s}\frac{X^{3}}{{g_{0}}^{2}} \end{array}$$

Noting that $X=x\cos\left(\omega t+\phi_{x}\right)$, the drive displacement results in parametric excitation of the sense mode through elastic modulation at frequency 2ω due to the term involving the mode-coupled cubic stiffness $\kappa_{0,4c,4s}$. Applying the averaging procedure to (21) with $Y=y\cos\left(\omega t+\phi_{x}+\phi_{yx}\right)$ gives the following when $\omega =\omega_{X}$:(22)$$\left[\begin{array}{cc} -{\omega_{X}}^{2}+{\omega_{Y}}^{2}+\lambda & -2\Gamma \omega_{X}\\ -2\Gamma \omega_{X} & {\omega_{X}}^{2}-{\omega_{Y}}^{2}+\lambda \end{array}\right]\bar{y}=\left[\begin{array}{c} f_{q}\\ f_{\Omega } \end{array}\right]$$
where $\bar{y}={\left[\begin{array}{cc} y\cos\phi_{yx} & y\sin\phi_{yx} \end{array}\right]}^{T}$ represents the phase-decomposed sense amplitude components. The sense amplitude component $y\cos\phi_{yx}$ describes the component of the sense response in phase/antiphase relative to the drive response, which is regarded as the rate output used for rate measurement. $y\sin\phi_{yx}$ describes the sense response component acting at a phase shift of ±π/2 relative to the drive response and is conventionally regarded as a quadrature output or error.

$\omega_{Y}$ is the effective sense frequency and λ describes the amplitude of the self-induced parametric excitation, given by the following:(23)$${\omega_{Y}}^{2}={\omega_{m}}^{2}-{\omega_{0,4c,4s}}^{2}+\kappa_{0,4c,4s}\frac{x^{2}}{2{g_{0}}^{2}}$$(24)$$\lambda =\kappa_{0,4c,4s}\frac{x^{2}}{4{g_{0}}^{2}}$$

On the right-hand side of (22), $f_{q}$ and $f_{\Omega }$ are the quadrature and Coriolis force amplitudes, respectively, given by the following:(25)$$f_{q}=\left\{\begin{array}{cc} -x{\omega_{m}}^{2}\Delta_{m}\sin4\Theta_{m}, & \;j=8\\ -\left(a_{4s}^{+}+a_{4s}^{-}\right)\gamma_{0,4s}\frac{3x^{3}}{4{g_{0}}^{2}}, & \;j=16 \end{array}\right.$$(26)$$f_{\Omega }=G_{\Omega }\Omega x\omega_{X}$$

When j=8, the quadrature force (25) is proportional to the drive amplitude and ring imperfection, and there is no dependence on the electrostatic tuning. However, when j=16, the quadrature force is independent of the imperfection due to satisfying (18), and is nonlinearly dependent on the drive amplitude. As such, the quadrature force magnitude is significantly reduced.

Solving (22) the sense amplitude components are as follows:(27)$$\bar{y}=\frac{1}{{\left(2\Gamma \omega_{X}\right)}^{2}+\left({\omega_{X}}^{2}-{\omega_{Y}}^{2}+\lambda \right)\left({\omega_{X}}^{2}-{\omega_{Y}}^{2}-\lambda \right)}\left[\begin{array}{c} -f_{\Omega }\left(2\Gamma \omega_{X}\right)-f_{q}\left({\omega_{X}}^{2}-{\omega_{Y}}^{2}+\lambda \right)\\ f_{\Omega }\left({\omega_{X}}^{2}-{\omega_{Y}}^{2}-\lambda \right)-f_{q}\left(2\Gamma \omega_{X}\right) \end{array}\right]$$

This equation indicates that when the drive frequency lies within the range ${\omega_{Y}}^{2}+\lambda <{\omega_{X}}^{2}<{\omega_{Y}}^{2}-\lambda$, parametric amplification occurs because electrostatic nonlinearity decreases the denominator and enhances the effective Q factor.

Using (20), (23), and (24), the following can be shown:(28)$${\omega_{X}}^{2}-{\omega_{Y}}^{2}-\lambda =\left[\gamma_{0,4c,4s}+\left(a_{4c}^{+}+a_{4c}^{-}\right)\gamma_{0,4c}-\kappa_{0,4c,4s}\right]\frac{3x^{2}}{4{g_{0}}^{2}}$$

This term is referred to as the nonlinear frequency imbalance and can be regarded as an extension of the linear frequency split when the parametric excitation amplitude λ is accounted for. Nonlinear frequency balance is achieved when ${\omega_{X}}^{2}-{\omega_{Y}}^{2}-\lambda =0$ and, as will be shown later, this has important consequences on linear electrostatic tuning.

For an ideal device operating at small drive amplitudes, $\omega_{X}=\omega_{Y}$, λ=0 and the dynamics are linear. For this case $\Delta_{m}=0$, $V_{4c}=V_{4s}=0$, so the sense amplitude components are as follows:(29)$${\bar{y}}_{t}=\left[\begin{array}{c} y_{t}\cos\phi_{yx,t}\\ y_{t}\sin\phi_{yx,t} \end{array}\right]=\left[-\begin{array}{c} \frac{f_{\Omega }}{2\Gamma \omega_{X}}\\ 0 \end{array}\right]$$

In (29) the ‘t’ subscript is used to emphasize that this is the ideal desired form that is “targeted” for a CVG device. The features of this ideal device output are:The rate output is proportional to the drive amplitude and angular rate;The quadrature output is zero, i.e., $\phi_{yx,t}=0$, ±π depending on the direction of rotation (sign of Ω).

Comparing (27) and (29), the ideal sense response is recaptured ($\bar{y}={\bar{y}}_{t}$) when the nonlinear frequency imbalance is nullified, i.e., ${\omega_{X}}^{2}-{\omega_{Y}}^{2}-\lambda =0$ and the quadrature force $f_{q}=0$. Substituting (11), (12), and (14) in (28), it can be shown that nonlinear frequency balancing is achieved for the continuous electrode case and for any electrode span when j=16, so these cases are well-suited to recapturing an ideal sense response.

In what follows, the rate sensitivity, bias rate, and quadrature error for sense amplitude components described by (27) are investigated and compared to the ideal device case described by (29).

### 4.3. Rate Sensitivity

To represent angular rate measurement, the rate output is represented in the form $y\cos\phi_{yx}=S\left(\Omega +\Omega_{0}\right)$. S is the rate sensitivity, given by the following:(30)$$S=\frac{G_{\Omega }x\omega_{X}\left(2\Gamma \omega_{X}\right)}{{\left(2\Gamma \omega_{X}\right)}^{2}+\left({\omega_{X}}^{2}-{\omega_{Y}}^{2}+\lambda \right)\left({\omega_{X}}^{2}-{\omega_{Y}}^{2}-\lambda \right)}$$

The rate sensitivity (30) generally does not increase proportionally with the drive amplitude because ${\omega_{X}}^{2}-{\omega_{Y}}^{2}+\lambda$ and nonlinear frequency imbalance ${\omega_{X}}^{2}-{\omega_{Y}}^{2}-\lambda$ depend nonlinearly on drive amplitude. To ensure the rate sensitivity approaches that of the ideal case, either nonlinear frequency balancing ${\omega_{X}}^{2}-{\omega_{Y}}^{2}-\lambda =0$ or ${\omega_{X}}^{2}-{\omega_{Y}}^{2}+\lambda =0$ must be satisfied. To illustrate these effects, the drive and sense frequencies and rate sensitivity for devices having j=8 and j=16 electrodes are investigated as the drive amplitude is increased. The device parameters and operating points used in the subsequent investigations are provided in Table 1.

For devices without linear electrostatic tuning $V_{4c}=V_{4s}=0$, whilst for devices with linear electrostatic tuning $V_{4c}=V_{4c}{|}_{t}$ and $V_{4s}=V_{4s}{|}_{t}$ to satisfy (17) and (18).

Figure 3a,b show the drive and sense resonant frequencies with and without linear electrostatic tuning implemented, respectively, for the j=8 case, whilst Figure 3c,d show the corresponding results for j=16. In these plots, the shaded region represents the range of sense frequencies bounded by self-induced parametric excitation, so any drive frequency lying in the shaded region parametrically amplifies the sense response, and for any drive frequencies outside this region, the sense response is parametrically diminished.

Figure 3a,c indicate that without linear electrostatic tuning the device has a linear frequency split of 9Hz. In contrast, Figure 3b,d include linear electrostatic tuning (i.e., $V_{4c}=V_{4c}{|}_{t}$ and $V_{4s}=V_{4s}{|}_{t}$) so the drive and sense frequencies are coincident at small drive amplitudes. In all cases the drive frequency reduces at a higher rate than the sense frequency as the drive amplitude increases, indicating the drive frequency has a stronger softening behavior.

For the 8-electrode case, Figure 3a indicates that the drive frequency lies within the frequency bounds of the self-induced parametric excitation when $6.6\%<x/g_{0}<12\%$ and nonlinear frequency balancing is achieved when $x/g_{0}=12\%$. Nonlinear frequency balancing does not occur in Figure 3b because the drive frequency $\omega_{X}$ is always less than frequency bound $\sqrt{{\omega_{Y}}^{2}+\lambda }$. The difference between $\omega_{X}$ and the lower frequency bound $\sqrt{{\omega_{Y}}^{2}+\lambda }$ increases monotonically as the drive amplitude increases, increasing the nonlinear frequency imbalance due to the higher drive softening rate.

For the 16-electrode case, Figure 3c,d indicate the drive frequency $\omega_{X}$ and the lower frequency bound $\sqrt{{\omega_{Y}}^{2}+\lambda }$ are approximately parallel, indicating the nonlinear frequency balance is approximately constant. This is because the single and mode-coupled cubic stiffness coefficients $\gamma_{0,4c,4s}$ and $\kappa_{0,4c,4s}$ are near identical due to the small electrostatic tuning voltages $V_{4c}$ and $V_{4s}$ (compare (11) and (14) for j=16 with $V_{b}=0$). In Figure 3c, the consequences of this are that nonlinear frequency balance cannot be achieved at all drive amplitudes and the drive frequency remains within the frequency bounds of the self-induced parametric excitation for drive amplitudes $x/g_{0}>7\%$. In contrast, Figure 3d indicates nonlinear frequency balancing is approximately achieved for all drive amplitudes ($\omega_{X}\approx \sqrt{{\omega_{Y}}^{2}+\lambda }$) when linear electrostatic tuning is achieved.

Having considered the drive and sense frequencies as the drive amplitude increases, the effect on rate sensitivity is considered next. In the results to be presented the theoretical results are compared to results obtained using the FE method. The FE results are obtained using the commercial package COMSOL Multiphysics version 6.2 [22] and are based on a two-dimensional ring model developed with the ‘Electrostatics’ interface. The FE model used is shown in Figure 4 below.

The ring model is divided into four domains. Domains 1 and 4 represent the vacuum dielectric between the ring and electrodes. Domains 2 and 3 represent the ring. Each domain is meshed with uniformly sized elements, featuring a circumferential resolution of 1.25° and a single element in the radial direction for each domain. A mesh convergence study was conducted to confirm that the mesh used ensures the accuracy of the results presented. Boundaries 1 and 2 denote the inner and outer electrodes, respectively. These boundaries are modeled as continuous circles for geometric simplicity, with voltages applied to them using piecewise functions dependent on angular position θ to reflect the voltage distributions applied. To model the eight supports, pairs of point forces in the radial and tangential directions are applied at eight uniformly spaced points on the ring centerline, where these forces are defined using linear functions of the radial and tangential displacements to represent linear radial and tangential springs. An angular rate Ω is applied to the ring model using the ‘Rotating Frame’ feature, followed by the initiation of a transient study that runs to obtain radial and tangential displacements of the ring once steady-state has been achieved. Radial and tangential displacement time histories are obtained at multiple points around the ring centerline, with a circumferential resolution of 3°. This is followed by a circumferential FFT of the radial and tangential displacements at each time step to extract the 2θ response, which yields the time histories of the drive and sense displacements, $X\left(t\right)$ and $Y\left(t\right)$. A temporal FFT is then evaluated on the time history of $Y\left(t\right)$ to obtain the frequency content. The rate and quadrature outputs are then derived from time histories corresponding to the frequency component at the drive frequency $\omega_{X}$. In the temporal FFT evaluation, only the final 10 load cycles of the time history are considered to ensure steady- state behavior. The FE results for the rate sensitivity are then obtained by linearly fitting the rate outputs at the angular rates specified in Table 1 using the relationship $y\cos\phi_{yx}=S\left(\Omega +\Omega_{0}\right)$, which also provides the bias rate $\Omega_{0}$.

Figure 5a,b show the drive amplitude dependencies of rate sensitivity S for cases involving 8 and 16 electrodes, respectively, with and without linear electrostatic tuning. The rate sensitivity is normalized with respect to the corresponding ideal form $S_{t}=-G_{\Omega }x/2\Gamma$ obtained from (30) with ${∆}_{m}=0$, λ=0 and low drive amplitudes.

For the 8-electrode case shown in Figure 5a, the rate sensitivity varies nonlinearly with drive amplitude, both with and without linear electrostatic tuning. The FE results are in agreement with the theoretical results and the small differences at large drive amplitudes are attributed to the presence of higher order electrostatic nonlinearities in the FE model. Without linear electrostatic tuning, the rate sensitivity is higher than the corresponding linear ideal form $S_{t}$ when the drive amplitude is between 6.6% and 12% of the capacitive gap. This increase is a consequence of the self-induced parametric amplification, as discussed in Figure 3a, which effectively increases the Q factor. When linear electrostatic tuning is implemented, the rate sensitivity matches that of the ideal case at small drive amplitudes but reduces as the drive amplitude increases. This is because the nonlinear frequency imbalance increases as the drive amplitude increases (see Figure 3b).

For the 16-electrode case shown in Figure 5b, the rate sensitivity increases monotonically as the drive amplitude increases both with and without linear electrostatic tuning. Without linear electrostatic tuning, the normalized rate sensitivity is greater than 1 for drive amplitudes greater than 7% of the capacitive gap. This is because the drive frequency remains within the frequency bounds of the self-induced parametric excitation (see Figure 3c). When linear electrostatic tuning is implemented, the nonlinear drive amplitude dependency is significantly suppressed and approximates the ideal case ($S/S_{t}\approx 1$) for the range of drive amplitudes considered. This occurs with a slight increase at larger drive amplitudes due to higher order electrostatic nonlinearities and nonlinear electrostatic forces from the drive voltage. This is because nonlinear frequency balancing is approximately achieved at all drive amplitudes shown (i.e., $\omega_{X}\approx \sqrt{{\omega_{Y}}^{2}+\lambda }$) (see Figure 3d). These results indicate the 16-electrode arrangement offers improved rate sensitivity linearity and trimming at larger drive amplitudes compared to the 8-electrode arrangement.

### 4.4. Bias Rate

Using the relationship $y\cos\phi_{yx}=S\left(\Omega +\Omega_{0}\right)$, the bias rate $\Omega_{0}$ is given by the following:(31)$$\Omega_{0}=\frac{f_{q}}{G_{\Omega }x\omega_{X}}\frac{{\omega_{X}}^{2}-{\omega_{Y}}^{2}+\lambda }{2\Gamma \omega_{X}}$$

The bias rate depends on the magnitude of the quadrature force $f_{q}$ and ${\omega_{X}}^{2}-{\omega_{Y}}^{2}+\lambda$ (which defines frequency bound $\omega_{X}=\sqrt{{\omega_{Y}}^{2}-\lambda }$ of the self-induced parametric excitation region, see Figure 3a–d). The magnitude of the quadrature force $f_{q}$ is much smaller for the 16-electrode case than for the 8-electrode case because the linear electrostatic tuning reduces the quadrature force to a residual cubic dependence on the drive amplitude for the 16-electrode case (see (25)). For the 8-electrode case, the linear elastic coupling remains uncompensated because $V_{4s}$ does not contribute to the modal dynamics, so $f_{q}$ remains directly dependent on the drive misalignment $\Theta_{m}$ and the imperfection magnitude $\Delta_{m}$.

The following investigates the effects of linear electrostatic tuning on bias rate for the 8- and 16-electrode arrangements. Figure 6a,b show how the bias rate varies with drive amplitude, with and without linear electrostatic tuning for the 8- and 16-electrode arrangements, respectively.

In Figure 6a,b, the imperfections induce a bias rate of −22°/s at small drive amplitudes for both electrode cases, without linear electrostatic tuning. Also, the FE results are in agreement with the theoretical results.

For the 8-electrode case, the bias rate is nullified at $x/g_{0}=6.6\%$ without linear electrostatic tuning when the drive frequency $\omega_{X}$ coincides with the frequency bound $\sqrt{{\omega_{Y}}^{2}-\lambda }$ (see Figure 3a). With linear electrostatic tuning, the bias rate is only nullified at small drive amplitudes and amplifies monotonically as the drive amplitude increases. This is because of the increasing difference between the drive frequency $\omega_{X}$ and the frequency bound $\sqrt{{\omega_{Y}}^{2}-\lambda }$ (see Figure 3b), and the quadrature force is uncompensated.

For the 16-electrode case, the bias rate amplifies monotonically as the drive amplitudes increases without linear electrostatic tuning and decreases slightly with linear electrostatic tuning. For drive amplitudes $x/g_{0}>7\%$, the difference between $\omega_{X}$ and frequency bound $\sqrt{{\omega_{Y}}^{2}-\lambda }$ (see Figure 3c) increases for the without linear electrostatic tuning case. Referring to Figure 5b, the rate sensitivity amplification achieved at higher drive amplitudes is accompanied by large bias rate errors if the imperfection-induced quadrature force is not addressed. When linear electrostatic tuning is implemented, the bias rate magnitude is significantly suppressed even at larger drive amplitudes due to the elimination of the linear elastic coupling. A slight negative trend is observed because the quadrature force magnitude $f_{q}>0$ for the case considered (see (25)).

### 4.5. Quadrature Error

The quadrature error is associated with a non-zero quadrature output represented by $y\sin\phi_{yx}$ in (27). As such, the quadrature error can be quantified in terms of the deviation of the relative phase $\phi_{yx}$ from that of the ideal case, where $\phi_{yx,t}=0$, ±π (see (29)). From (27), the relative phase is given by the following:(32)$$\phi_{yx}=\tan^{-1}\frac{f_{\Omega }\left({\omega_{X}}^{2}-{\omega_{Y}}^{2}-\lambda \right)-f_{q}\left(2\Gamma \omega_{X}\right)}{-f_{\Omega }\left(2\Gamma \omega_{X}\right)-f_{q}\left({\omega_{X}}^{2}-{\omega_{Y}}^{2}+\lambda \right)}$$

The quadrature output is generally nullified only if nonlinear frequency balancing applies (${\omega_{X}}^{2}-{\omega_{Y}}^{2}=\lambda$) and the quadrature force is nullified ($f_{q}=0$). Another important trait of the relative phase in linear operation is that the relative phase does not vary with drive amplitude. However, this is not the case in the presence of electrostatic nonlinearities due to the quadratic drive amplitude dependency of ${\omega_{X}}^{2}$, ${\omega_{Y}}^{2}$ and λ.

The following investigates the effects of linear electrostatic tuning on the relative phase for the 8- and 16-electrode arrangements. Figure 7a,b show the drive amplitude dependency of the relative phase with and without linear electrostatic tuning for both cases with an angular rate of Ω=250°/s. In both cases the FE results are in good agreement with the theoretical results.

At small drive amplitudes, the relative phase is insensitive to drive amplitude variations regardless of the implementation of linear electrostatic tuning, indicating the sense response varies linearly at these drive amplitudes.

For the 8-electrode case, Figure 7a shows that the relative phase does not approach 180° regardless of the presence of the linear electrostatic tuning. This is because the quadrature force is completely uncompensated (see (25)).

For the 16-electrode case, Figure 7b shows that linear electrostatic tuning significantly reduces the quadrature error as the relative phase remains approximately at −180°, even at increased drive amplitudes. This demonstrates increased effectiveness of linear electrostatic tuning for quadrature error performance when using a 16-electrode arrangement compared to an 8-electrode arrangement. The relative phase is insensitive to drive amplitude variations even at increased drive amplitudes for two reasons. First, the quadrature force is reduced to a residual nonlinear form (see (25) for j=16). Secondly, nonlinear frequency balancing is achieved, i.e., $\omega_{X}\approx \sqrt{{\omega_{Y}}^{2}+\lambda }$ as is shown in Figure 3d.

The results in Figure 5, Figure 6 and Figure 7 indicate linear electrostatic tuning is significantly more effective for recapturing ideal rate-sensing performance in the 16-electrode arrangement than the 8-electrode arrangement. This is attributed to how the nonlinear frequency imbalance and quadrature force change with drive amplitude. For the 8-electrode case the nonlinear frequency imbalance changes significantly with drive amplitude causing the rate sensitivity to degrade at larger drive amplitudes, and the bias rates and quadrature errors vary nonlinearly with drive amplitude. In contrast, for the 16-electrode case the nonlinear frequency imbalance is near constant with drive amplitude, ensuring the rate sensitivity is approximately equal to that of an idealized device. The quadrature force is also significantly reduced which helps to minimize bias rate and quadrature errors.

## 5. Closed Loop Sense Force Balancing

In this section, closed loop sense force balancing is investigated. The key difference compared to the open loop linear electrostatic tuning (see Section 4) is that the closed loop sense force balancing is not aimed at modifying the elastic properties of the drive and sense modes. Instead, the sense mode is excited directly, similar to FTR mode operation [1,23]. However, in contrast to FTR, the sense amplitude is not nullified. That is, the forcing conditions involve ensuring the sense amplitude components are $\bar{y}={\bar{y}}_{t}$ (see (29)) instead of $\bar{y}=0$. Additionally, this also has the benefit of allowing rate measurements to be made using either the balancing voltage or the sense response.

In this section, the linear electrostatic tuning voltage components in (2a) and (2b) are set to zero in the general governing equations of motion (7a) and (7b) and the resulting equations are used to determine the balancing voltage conditions needed to counteract the effects of imperfection and nonlinear force imbalance to recapture ideal sense response. Similar to Section 4, the effectiveness of this scheme is assessed by considering the rate sensitivity, bias rate, and quadrature error for 8- and 16-electrode arrangements. Based on the balancing voltage conditions to recapture ideal sense response, the use of the balancing voltage to measure angular rate for rate sensing is then investigated. The resulting rate-sensing scheme is similarly assessed for the 8- and 16-electrode arrangements.

### 5.1. Determining Balancing Voltage Conditions

Setting the linear electrostatic tuning voltage components to zero ($a_{4c}^{\pm }V_{4c}=a_{4s}^{\pm }V_{4s}=0$) in (7a) and (7b) and applying the same assumptions used in Section 4.2 gives the following:(33)$$\ddot{X}+2\Gamma \dot{X}+\left[{\omega_{m}}^{2}\left(1+\Delta_{m}\cos4\Theta_{m}\right)-{\omega_{0,b}}^{2}\right]X+\gamma_{0,b}\frac{X^{3}}{{g_{0}}^{2}}=\left(a_{AC}^{+}-a_{AC}^{-}\right)\chi_{0,AC}\cos\omega t\left(1+c_{X^{3}}\frac{X^{2}}{{g_{0}}^{2}}\right)$$(34)$$\begin{array}{ll} \ddot{Y}+2\Gamma \dot{Y} & +\left[{\omega_{m}}^{2}\left(1-\Delta_{m}\cos4\Theta_{m}\right)-{\omega_{0,b}}^{2}+\kappa_{0,b}\frac{X^{2}}{{g_{0}}^{2}}\right]Y\\ & \;\;\;=-G_{\Omega }\Omega \dot{X}-{\omega_{m}}^{2}\Delta_{m}\sin4\Theta_{m}X+\chi_{b}\cos\left(\omega t+\phi_{x}+i_{b}\pi \right)\left(1+c_{X^{2}Y}\frac{X^{2}}{{g_{0}}^{2}}\right) \end{array}$$
where ${\omega_{0,b}}^{2}$, $\kappa_{0,b}$, $\gamma_{0,b}$ are simplified versions of ${\omega_{0,4c,4s,b}}^{2}$, $\kappa_{0,4c,4s,b}$, $\gamma_{0,4c,4s,b}$, respectively.

Following an identical approach to that in Section 4.2, applying the averaging procedure on (34) gives the following when $\omega =\omega_{X}$ for drive resonance:(35)$$\left[\begin{array}{cc} -{\omega_{X}}^{2}+{\omega_{Y}}^{2}+\lambda & -2\Gamma \omega_{X}\\ -2\Gamma \omega_{X} & {\omega_{X}}^{2}-{\omega_{Y}}^{2}+\lambda \end{array}\right]\bar{y}=\left[\begin{array}{c} f_{q,m}+f_{q,\chi }\\ f_{\Omega } \end{array}\right]$$
where the drive frequency $\omega_{X}$, sense frequency $\omega_{Y}$ and parametric excitation amplitude λ are given by the following:(36)$${\omega_{X}}^{2}={\omega_{m}}^{2}\left(1+\Delta_{m}\cos4\Theta_{m}\right)-{\omega_{0,b}}^{2}+\gamma_{0,b}\frac{3x^{2}}{4{g_{0}}^{2}}$$(37)$${\omega_{Y}}^{2}={\omega_{m}}^{2}\left(1-\Delta_{m}\cos4\Theta_{m}\right)-{\omega_{0,b}}^{2}+\kappa_{0,b}\frac{x^{2}}{2{g_{0}}^{2}}$$(38)$$\lambda =\kappa_{0,b}\frac{x^{2}}{4{g_{0}}^{2}}$$
and $f_{q,m}$ and $f_{q,\chi }$ are components of the quadrature force amplitude associated with the mechanical imperfection and balancing voltage, respectively, given by the following:(39)$$f_{q,m}=-x{\omega_{m}}^{2}\Delta_{m}\sin4\Theta_{m}$$(40)$$f_{q,\chi }=\chi_{b}\cos i_{b}\pi \left(1+c_{X^{2}Y}\frac{3x^{2}}{4{g_{0}}^{2}}\right)$$

From (35), the balancing force does not interact with the Coriolis force as these forces are phase decoupled. This is the main difference of the present balancing approach compared to FTR mode operation.

To achieve ideal sense response, the solution to (35) must be $\bar{y}={\bar{y}}_{t}$ (see (29)). Substituting $\bar{y}={\bar{y}}_{t}$ into (35) and solving for $f_{q,\chi }$, the balancing force amplitude $f_{q,\chi }$ to achieve ideal sense response is as follows:(41)$$f_{q,\chi }=f_{q,\chi }{|}_{t}=-f_{q,m}+\left({\omega_{X}}^{2}-{\omega_{Y}}^{2}-\lambda \right)\frac{G_{\Omega }\Omega x}{2\Gamma }$$

Using (A3) in (40) and substituting the resulting expression into (41), the required balancing voltage amplitude $V_{b}$ and phase index $i_{b}$ are given by the following:(42)$${\left(-1\right)}^{i_{b}{|}_{t}}{V_{b}{|}_{t}}^{2}=-\frac{f_{q,m}-\left({\omega_{X}}^{2}-{\omega_{Y}}^{2}-\lambda \right)\frac{G_{\Omega }\Omega x}{2\Gamma }}{l_{b}\left(1+c_{X^{2}Y}\frac{3x^{2}}{4{g_{0}}^{2}}\right)}$$
where the relationship $\cos i_{b}{|}_{t}\pi ={\left(-1\right)}^{i_{b}{|}_{t}}$ has been used, since the balancing force phase index $i_{b}$ is an integer, and $l_{b}$ is given in (A9).

On the right side of (41), the net force imbalance arises from the quadrature force due to imperfection ($f_{q,m}$) and nonlinear frequency imbalance ${\omega_{X}}^{2}-{\omega_{Y}}^{2}-\lambda$. The sign of the total force imbalance on the right side of (41) dictates the required parity of the phase index $i_{b}{|}_{t}$ in (42). In practice, however, the various terms on the right side of (41) are not necessarily known. Since the ideal sense response has zero quadrature output (see (29)), the required balancing voltage amplitude $V_{b}$ and phase index $i_{b}$ to satisfy (42) must be selected so the relative phase is 0 or ±π. In practice, this would require the implementation of a control loop to monitor the relative phase to identify the correct combination of $V_{b}$ and $i_{b}$. This is particularly important because the required $V_{b}$ and $i_{b}$ in (42) vary with drive amplitude and angular rate. Due to the dependence of this approach on quadrature response detection, in practical cases, it is anticipated that the robustness of this approach is dictated by noise specifically within the pickoff circuitry [24] of the quadrature output of the device.

To illustrate the effects of implementing the balancing voltage, the drive and sense frequencies are considered, and the resulting rate sensitivity, bias rate, and quadrature error are investigated for the 8- and 16-electrode arrangements. The devices considered have the parameters and operating conditions listed in Table 1, except that the constants in the voltage distribution in (2a) and (2b) are modified to $a_{AC}^{\pm }=\pm 1$, and $a_{4c}^{\pm }=a_{4s}^{\pm }=0$ because linear electrostatic tuning is not implemented.

Figure 8a,b show the drive amplitude dependencies of the drive and sense resonant frequencies for the 8- and 16-electrode arrangements, respectively, with ($V_{b}=V_{b}{|}_{t}$) and without ($V_{b}=0$) sense force balancing. The shaded regions represent the frequency bounds of the self-induced parametric excitation, similar to the results in Figure 3a–d. For the results including sense force balancing, the required balancing voltage amplitude $V_{b}{|}_{t}$ is calculated using (42) at each drive amplitude considered and the balancing voltage phase index $i_{b}$ is set at 0 or 1 depending on the sign of the right side of (41). The results without sense force balancing are identical to the results in Figure 3a,c without the linear electrostatic tuning, as the same control systems are considered.

Figure 8a,b show that the balancing voltage has negligible impact on the drive and sense frequencies and the self-induced parametric excitation. This is because the balancing voltage amplitude is much smaller than the bias voltage, and so does not significantly affect the modal properties.

Figure 9a,b compare the relative phase for the 8- and 16-electrode arrangements, respectively, as the drive amplitude is increased with ($V_{b}=V_{b}{|}_{t}$) and without ($V_{b}=0$) the balancing voltage when the angular rate is Ω=250°/s. Similar to the results in Section 4.3 to 4.5, theoretical results are validated against numerical results from FE transient studies. For the FE results, $V_{b}{|}_{t}$ is determined by manually adjusting the balancing voltage amplitude $V_{b}$ to nullify the quadrature output until a relative phase tolerance of ±0.1° is achieved.

Figure 9a,b show that the balancing voltage is capable of nullifying the quadrature error at any given drive amplitude, for both j=8 and j=16. These results highlight a distinct advantage of the sense force balancing approach compared to the linear electrostatic tuning case. The balancing force counteracts both the imperfection-induced linear elastic coupling and nonlinear frequency imbalance, which are not generally eliminated using linear electrostatic tuning.

Figure 10a,b show the corresponding results for the normalized rate sensitivity similar to Figure 5a,b, and Figure 11a,b show the corresponding results for the bias rate $\Omega_{0}$. When the balancing voltage $V_{b}=V_{b}{|}_{t}$ the aim is to achieve ideal performance with normalized rate sensitivity of 1 ($S=S_{t}$) and the bias rate $\Omega_{0}=0$.

In Figure 10a,b, the balancing force significantly suppresses the nonlinear drive amplitude dependency of the rate sensitivity and the ideal rate sensitivity is closely reproduced. However, there is a slight increase in rate sensitivity at larger drive amplitudes relative to the linear, ideal case $S_{t}$. This is due to the presence of small parametric amplification stemming from nonlinear electrostatic forces associated with the drive voltage, which has been neglected in the present analysis.

In Figure 11a,b, the balancing force significantly reduces the bias rate magnitude and suppresses the drive amplitude sensitivity of the bias rate as the drive amplitude increases. The bias rate magnitude is small, even at increased drive amplitudes, allowing the rate output to be linearly amplified at larger drive amplitudes without the introduction of bias rate errors.

The results confirm the effectiveness of implementing a balancing force to replicate ideal sense response in the presence of imperfections and electrostatic nonlinearities. The ability to achieve this at increased drive amplitudes relies on actively adjusting the balancing voltage with changes in drive amplitude and angular rate (see (42)) and is a distinct advantage of this scheme compared to linear electrostatic tuning. As the required balancing voltage to achieve force balance changes with angular rate, the balancing voltage itself can be used as a means of measuring angular rate. This is investigated next.

### 5.2. Using Balancing Voltage to Measure Angular Rate

Equation (42) indicates that when the balancing voltage has been set to nullify the quadrature output, ${\left(-1\right)}^{i_{b}{|}_{t}}{V_{b}{|}_{t}}^{2}$ is linearly related to the angular rate to be measured. A consequence of this is that the angular rate can be determined directly from the balancing voltage. To investigate the effectiveness of using the balancing voltage to measure angular rate, (42) is expressed in the form ${\left(-1\right)}^{i_{b}}{V_{b}{|}_{t}}^{2}=S_{bv}\left(\Omega +\Omega_{0,bv}\right)$, where $S_{bv}$ and $\Omega_{0,bv}$ are the rate sensitivity and bias rate, respectively, associated with the balancing voltage. Using (39) in (42), the following can be shown:(43)$$S_{bv}=\frac{\frac{G_{\Omega }x}{2\Gamma }\left({\omega_{X}}^{2}-{\omega_{Y}}^{2}-\lambda \right)}{l_{b}\left(1+c_{X^{2}Y}\frac{3x^{2}}{4{g_{0}}^{2}}\right)}$$(44)$$\Omega_{0,bv}=\frac{{\omega_{m}}^{2}\Delta_{m}\sin4\Theta_{m}}{\frac{G_{\Omega }}{2\Gamma }\left({\omega_{X}}^{2}-{\omega_{Y}}^{2}-\lambda \right)}$$

(43) and (44) indicate the rate sensitivity increases, and the bias rate decreases as the nonlinear frequency imbalance ${\omega_{X}}^{2}-{\omega_{Y}}^{2}-\lambda$ increases and damping Γ decreases. For the special cases when j=8, δ=π/4 or j=16 regardless of electrode arrangement, it can be shown that ${\omega_{X}}^{2}-{\omega_{Y}}^{2}-\lambda$ is constant and does not vary with drive amplitude. In these cases, $S_{bv}$ is approximately proportional to drive amplitude and $\Omega_{0,bv}$ is constant with drive amplitude.

The following numerical results illustrate the rate sensitivity and bias rate associated with using the balancing voltage to measure angular rate. Figure 12a,b and Figure 13a,b show how the rate sensitivity and bias rate vary with drive amplitude for the 8- and 16-electrode arrangements studied previously. However, reduced damping cases of Γ=5.65 Hz and 0.565 Hz are also included here, corresponding to Q factors of 1200 and 12000, respectively. FE results are provided to validate the theoretical results calculated using (43) and (44), but only for the control damping case in Table 1. This is due to the prohibitively long computation durations required to achieve steady-state behavior within the transient studies in COMSOL. For the control damping case, the compute time is in the order of days on a HPC. In all cases, the obtained FE results are in agreement with the theoretical results.

For the 8-electrode arrangement, Figure 12a shows that the rate sensitivity generally varies nonlinearly with drive amplitude. At low drive amplitudes, the nonlinear frequency imbalance ${\omega_{X}}^{2}-{\omega_{Y}}^{2}-\lambda$ is negligible, so the rate sensitivity increases in proportion with drive amplitude due to the linear amplification of the Coriolis force amplitude $f_{\Omega }$. As the drive amplitude increases the effects of the nonlinear frequency imbalance become increasingly significant. For $x/g_{0}\leq 0.12$, the nonlinear frequency imbalance degrades the rate sensitivity, and at $x/g_{0}=0.12$ the rate sensitivity is nullified because nonlinear frequency balancing is achieved (see Figure 8a). For $x/g_{0}>0.12$, the rate sensitivity increases nonlinearly with drive amplitude, but with opposite polarity. Reducing the damping amplifies the rate sensitivity as the rate sensitivity is inversely proportional to the damping (see (43)), but the nature of the drive amplitude dependence is unchanged. This shows that for the 8-electrode arrangement, large rate sensitivities can be achieved at large drive amplitudes far from the nonlinear frequency balancing point, particularly if the damping is also small.

For the 16-electrode arrangement, Figure 12b shows that the rate sensitivity increases linearly with drive amplitude because the nonlinear frequency imbalance is constant (see Figure 8b). As such, unlike the 8-electrode arrangement, the rate sensitivity does not deteriorate at specific drive amplitudes. Reducing the damping similarly amplifies the rate sensitivity. As such, large drive amplitudes and minimal damping are beneficial for both 8- and 16-electrode arrangements. However, the nonlinear rate sensitivity amplification for the 8-electrode arrangement is more significant.

In Figure 13a, for all damping levels considered, the bias rate increases as the drive amplitude increases up to $x/g_{0}=0.12$ and reduces with opposite polarity as the drive amplitude increases above $x/g_{0}=0.12$. The large bias rates near drive amplitudes $x/g_{0}\approx 0.12$ are due to the nonlinear frequency balancing as shown in Figure 8a, so these drive amplitudes should be avoided if the balancing voltage is used to sense rate. Reducing the damping significantly reduces the bias rate at all drive amplitudes, as the bias rate is proportional to the damping (see (44)). In addition, the range of drive amplitudes with amplified bias rate is significantly narrowed when the damping is reduced.

In Figure 13b, for the 16-electrode arrangement, the bias rate is constant with drive amplitude because the nonlinear frequency imbalance is invariant with drive amplitude for this case. This contrasts the 8-electrode arrangement in Figure 13a and highlights the benefit of using 16 electrodes as the bias rate does not degrade at specific drive amplitudes. For the control damping case (Γ=56.5 Hz), the bias rate is large (~900°/s) due the large linear damping coefficient Γ. However, the bias rate is uniformly reduced at all drive amplitudes when the damping is reduced.

The results in Figure 12a,b and Figure 13a,b conclude that the 8-electrode arrangement generally offers better rate-sensing performance than the 16-electrode arrangement, owing to the nonlinear rate sensitivity amplification at large drive amplitudes. The only exception to this is at drive amplitudes close to nonlinear frequency balancing. However, this can be mitigated with reduced damping. For the specific cases of minimal damping and large nonlinear frequency imbalances, the balancing voltage is a viable alternative to the corrected sense response to measure angular rate.

## 6. Conclusions

A mathematical model has been developed to investigate the effects of linear electrostatic tuning and sense force balancing on the rate-sensing performance of imperfect ring-based CVGs having 8- and 16-evenly distributed electrodes, where rate-sensing performance is assessed in terms of rate sensitivity, bias rate, and quadrature error.

Linear electrostatic tuning provides an effective means of recapturing ideal rate-sensing performance at low drive amplitudes. However, as the drive amplitude increases, the rate-sensing performance for 8- and 16-electrode arrangements exhibits differing susceptibilities to electrostatic nonlinearities depending on the nonlinear frequency imbalance. For the 8-electrode arrangement, the linear elastic coupling and quadrature force cannot be negated, so the rate sensitivity, bias rate, and quadrature error vary nonlinearly with the drive amplitude. The 16-electrode arrangement offers improved rate-sensing performance linearity because the nonlinear frequency imbalance is relatively insensitive to drive amplitude. This enables ideal rate-sensing performance to be approximated even at larger drive amplitudes. However, at larger drive amplitudes, the voltages used for linear electrostatic tuning introduce additional nonlinear elastic coupling, which slightly modify the rate sensitivity, bias rate, and quadrature error. As such, for both 8- and 16-electrode arrangements, the electrostatic nonlinearities have degrading effects on the effectiveness of linear electrostatic tuning, albeit to different extents. In all cases, the 16-electrode arrangement offers better performance when linear electrostatic tuning is implemented.

Sense force balancing is implemented by applying direct forces to the sense mode aimed at eliminating the quadrature output. It is shown that this scheme allows ideal sense response to be recaptured, even at increased drive amplitudes, for both 8- and 16-electrode arrangements. The ability to recapture ideal rate-sensing performance with the 8-electrode arrangement is particularly attractive, as the 8-electrode arrangement offers better manufacturing simplicity. To implement this scheme in practice, it is necessary to use control loops to adjust the balancing voltage amplitude and phase to nullify the quadrature output which depends on angular rate and drive amplitude. Rate sensing can then be achieved either using the recaptured ideal sense response or the balancing voltage, which is rate dependent. When using the balancing voltage for rate measurement, the 8-electrode arrangement offers better rate sensitivity and bias rate performance than the 16-electrode arrangement, particularly at large amplitudes, except for a narrow range of drive amplitudes where nonlinear frequency balancing occurs. The balancing voltage is particularly well-suited for measuring angular rate for the specific cases of large nonlinear frequency imbalances and minimal damping, as these amplify the balancing voltage rate sensitivity and suppresses bias rate.

## Figures and Tables

**Figure 1 sensors-25-02263-f001:**
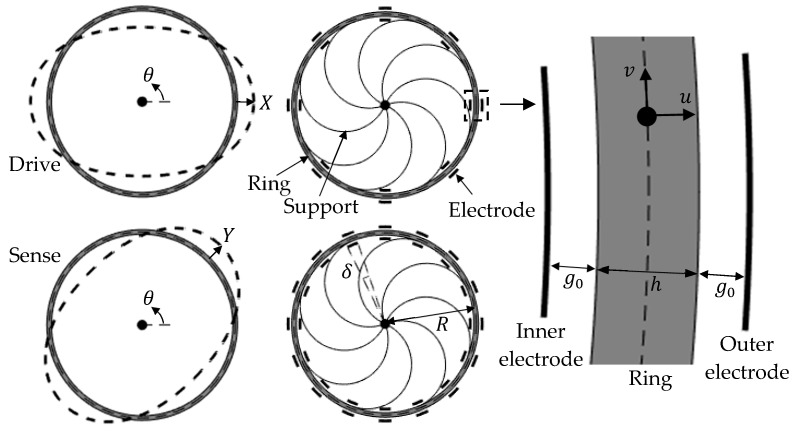
Device setup with 8 and 16 evenly-distributed inner and outer electrodes.

**Figure 2 sensors-25-02263-f002:**
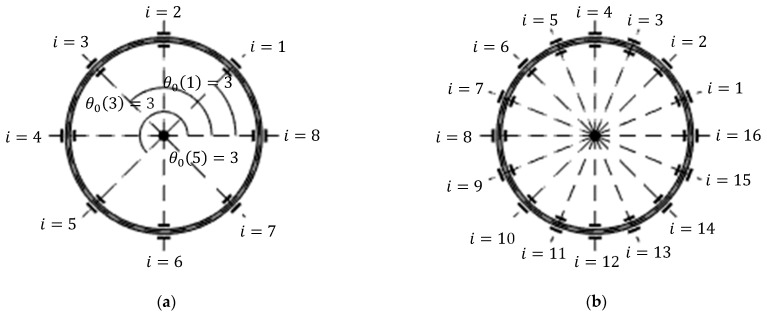
Identification of each inner/outer electrode using index i for (a) j=8 and (b) j=16.

**Figure 3 sensors-25-02263-f003:**
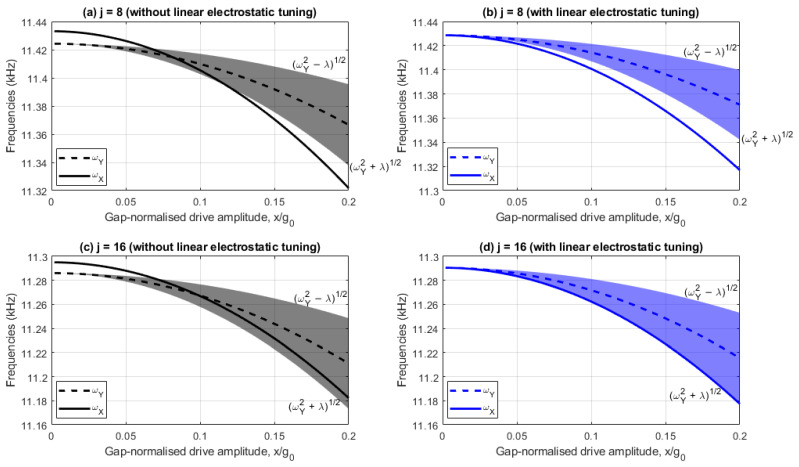
Drive and sense frequencies for (**a**) $j=8,V_{4c}=0,V_{4s}=0$, (**b**) $j=8,V_{4c}=V_{4c}{|}_{t},V_{4s}=V_{4s}{|}_{t}$, (**c**) $j=16,V_{4c}=0,V_{4s}=0$, and (**d**) $j=16,V_{4c}=V_{4c}{|}_{t},V_{4s}=V_{4s}{|}_{t}$ with frequency bounds of self-induced parametric excitation.

**Figure 4 sensors-25-02263-f004:**
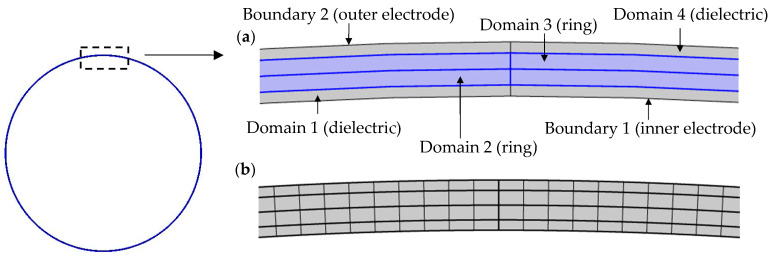
Two-dimensional FE ring model showing (**a**) the main domains and boundaries and (**b**) mesh elements.

**Figure 5 sensors-25-02263-f005:**
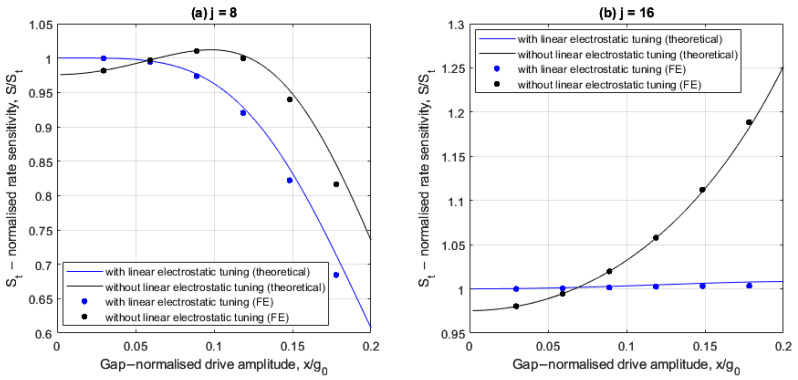
Rate sensitivity relative to the linear, ideal case for the (**a**) 8- and (**b**) 16-electrode arrangement, with and without linear electrostatic tuning.

**Figure 6 sensors-25-02263-f006:**
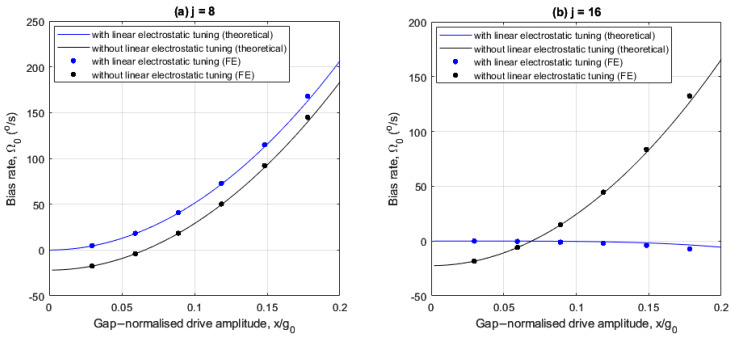
Drive amplitude dependency of the bias rate for the (**a**) 8- and (**b**) 16- electrode arrangements with and without linear electrostatic tuning.

**Figure 7 sensors-25-02263-f007:**
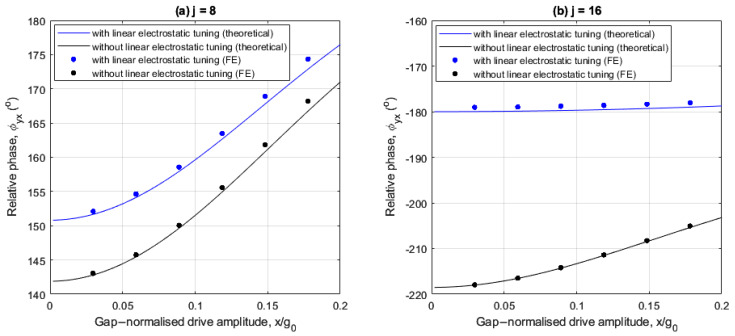
Drive amplitude dependency of the relative phase for (**a**) 8- and (**b**) 16-electrode arrangements with and without linear electrostatic tuning.

**Figure 8 sensors-25-02263-f008:**
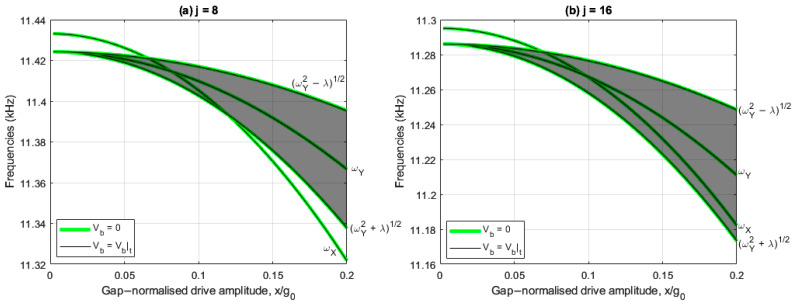
Comparison of drive and sense resonant frequencies with self-induced parametric excitation frequency bounds for (**a**) 8- and (**b**) 16-electrode arrangements, with and without force balancing.

**Figure 9 sensors-25-02263-f009:**
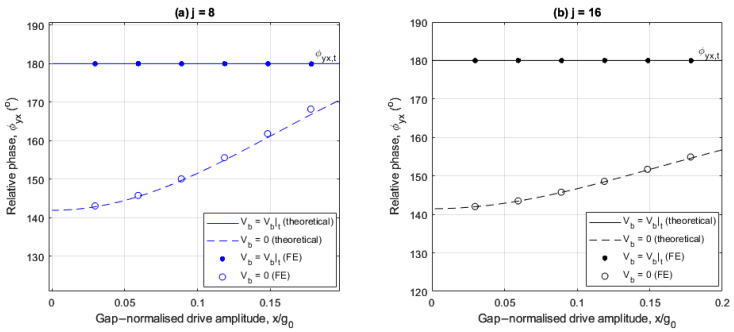
Effect of balancing voltage implementation on the relative phase for (**a**) 8- and (**b**) 16-electrode arrangements.

**Figure 10 sensors-25-02263-f010:**
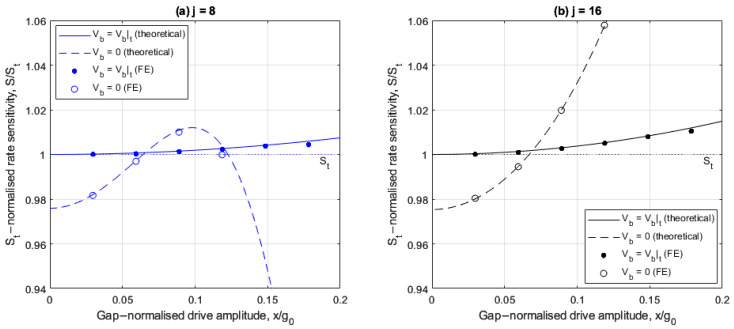
Effect of balancing voltage implementation on the rate sensitivity compared to the linear, ideal case for (**a**) 8- and (**b**) 16-electrode arrangements.

**Figure 11 sensors-25-02263-f011:**
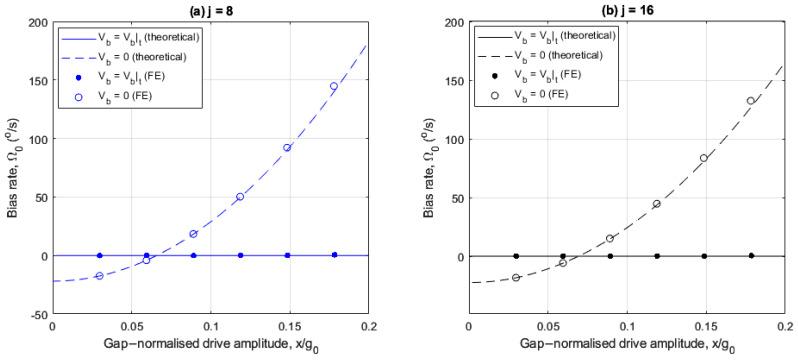
Effect of balancing voltage implementation on the bias rate for (**a**) 8- and (**b**) 16-electrode arrangements.

**Figure 12 sensors-25-02263-f012:**
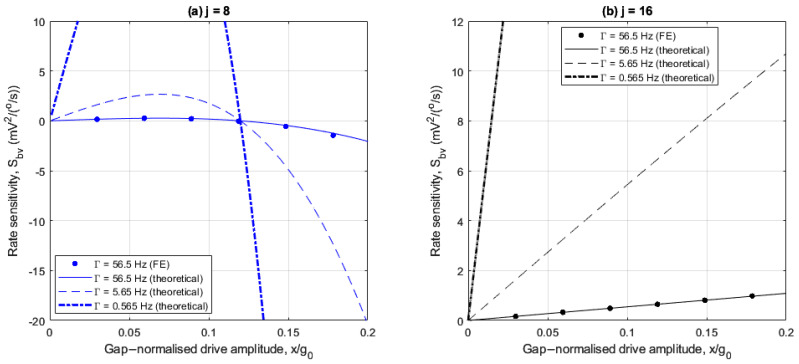
Drive amplitude dependence of balancing voltage rate sensitivity for (**a**) 8- and (**b**) 16-electrode arrangements.

**Figure 13 sensors-25-02263-f013:**
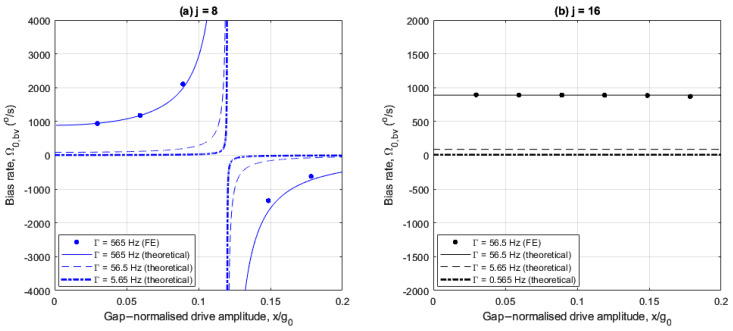
Drive amplitude dependence of balancing voltage bias rate for (**a**) 8- and (**b**) 16-electrode arrangements.

**Table 1 sensors-25-02263-t001:** Parameters and operating points for devices with linear electrostatic tuning implemented.

Parameters/Operating Points	Value	
$\omega_{m}(kHz)$	13.558	
$\Delta_{m}$	5.5 × 10^−4^	
$\Theta_{m}\left({}^{\circ}\right)$	1	
$\left(a_{AC}^{+},a_{AC}^{-}\right)$	(1, −2)	
$\left(a_{4c}^{+},a_{4c}^{-}\right)$	(1, −0.5)	
$\left(a_{4s}^{+},a_{4s}^{-}\right)$	(1, −0.5)	
$\Gamma \left(Hz\right)$	56.5	
$\Omega \left({}^{\circ}/s\right)$	230–250	
$\rho \left(kg/m^{3}\right)$	2320	
$h\left(\mu m\right)$	4	
$g_{0}\left(\mu m\right)$	1.4	
j	8	16
$\delta \left({}^{\circ}\right)$	38	19
$V_{0}\left(V\right)$	2.11	2.18
$V_{4c}{|}_{t}\left(mV\right)$	11	16.9
$V_{4s}{|}_{t}\left(mV\right)$	N/A	1.2

## Data Availability

The original contributions presented in this study are included in the article. Further inquiries can be directed to the corresponding authors.

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
