# Peer review of "Rate-Sensing Performance of Imperfect Capacitive Ring-Based MEMS Coriolis Vibrating Gyroscopes at Large Drive Amplitudesâ€"

_sensors, 2025, doi:10.3390/s25072263_

Round 1
Reviewer 1 Report
Comments and Suggestions for Authors
I appreciate the authors submission of the interesting paper "Rate sensing performance of imperfect capacitive ring-based MEMS Coriolis Vibrating Gyroscopes at large drive amplitudes." The manuscript discusses the impact of nonlinear electrostatic effects on MEMS vibrating gyroscopes. The work presents a comprehensive mathematical model along with extensive simulation analyses. Nonetheless, multiple factors require attention before endorsing acceptance:
- The introduction is organised effectively; however, the authors reference prior work that was presented at a conference (line 92). It would be beneficial to explicitly outline the additional content or analysis included in this extended version compared to what was shared at the conference [14].
- Furthermore, it is essential to enhance the introduction section by incorporating additional relevant references, especially related to vibrating ring gyroscopes. Below are a few recommendations for consideration.
- Hu Z, Gallacher BJ. Effects of nonlinearity on the angular drift error of an electrostatic MEMS rate integrating gyroscope. IEEE Sensors Journal. 2019 Jul 16;19(22):10271-80.
2. Braghin F, Resta F, Leo E, Spinola G. Nonlinear dynamics of vibrating MEMS. Sensors and Actuators A: Physical. 2007 Feb 28;134(1):98-108.
3. Gill WA, Howard I, Mazhar I, McKee K. A Detailed Analysis of the Dynamic Behavior of a MEMS Vibrating Internal Ring Gyroscope. Micromachines. 2024 Aug 30;15(9):1107.
- The authors mention 8 evenly-spaced supports (line 113), but these are not clearly illustrated in Figures 1 and 2. Including a diagram explicitly showing these supports would improve clarity.
- The mathematical derivations presented in equations (7a) and (7b) are thorough and comprehensive. It would be advantageous to briefly outline the physical significance of the critical nonlinear terms regarding device functionality. Clearly defining the physical significance of these nonlinear terms may enhance the reader's understanding of the derivations.
- The manuscript examines closed-loop sense force balancing as a means to enhance optimal rate sensing performance. Clearly establish the ways in which practical execution can vary from theoretical assumptions. It is essential to take into account any possible real-world limitations, including factors like noise or microfabrication tolerances.
- Double-check consistency in formatting equations, especially equations (7a, 7b, 19-24). Ensure uniform use of parentheses, subscript/superscript conventions, and clearly defined symbols.
The manuscript presents valuable contributions to the field and clearly demonstrates the potential benefits of the proposed electrostatic tuning and sense force balancing methods. However, addressing the above comments will significantly enhance the manuscript's clarity and impact.
Author Response
Comments |
Response |
The introduction is organised effectively; however, the authors reference prior work that was presented at a conference (line 92). It would be beneficial to explicitly outline the additional content or analysis included in this extended version compared to what was shared at the conference [14]. |
Reference to the conference paper [16] has been added the revised manuscript. Lines 99-102 on page 3 have been added to outline the distinctive contribution of the paper compared to what was shared at the conference. |
Furthermore, it is essential to enhance the introduction section by incorporating additional relevant references, especially related to vibrating ring gyroscopes. Below are a few recommendations for consideration. 1. Hu Z, Gallacher BJ. Effects of nonlinearity on the angular drift error of an electrostatic MEMS rate integrating gyroscope. IEEE Sensors Journal. 2019 Jul 16;19(22):10271-80. |
Lines 64-69 on page 2 of the revised manuscript have been added to include reference to previous work on quadrature nulling and the effects of electrostatic nonlinearity on its implementation, albeit on a rate integrating gyroscope rather than rate mode which is the focus of this paper. This is the first paper within the recommended list. |
The authors mention 8 evenly-spaced supports (line 113), but these are not clearly illustrated in Figures 1 and 2. Including a diagram explicitly showing these supports would improve clarity. |
The 8 evenly-spaced supports are now included in Figure 1. |
The mathematical derivations presented in equations (7a) and (7b) are thorough and comprehensive. It would be advantageous to briefly outline the physical significance of the critical nonlinear terms regarding device functionality. Clearly defining the physical significance of these nonlinear terms may enhance the reader's understanding of the derivations. |
The paragraph extending from lines 223 to 229 on page 8 has been added to define the physical significance of the dominant nonlinear terms and brief explanations of the main effects which are demonstrated in later sections. |
The manuscript examines closed-loop sense force balancing as a means to enhance optimal rate sensing performance. Clearly establish the ways in which practical execution can vary from theoretical assumptions. It is essential to take into account any possible real-world limitations, including factors like noise or microfabrication tolerances. |
The anticipated practical limitation is discussed in lines 563 – 566 on page 18. |
Double-check consistency in formatting equations, especially equations (7a, 7b, 19-24). Ensure uniform use of parentheses, subscript/superscript conventions, and clearly defined symbols. |
The use of the parentheses has been applied consistently in these equations with the ‘()’, ‘[]’ parentheses applied in order from inner to outer nestings. The use of the subscript/superscript have all been explicitly defined. The ‘+/-‘ superscripts are defined within line 150 on page 5 while the use of the various subscripts associated with the voltage components have been explained within lines 210 - 212 on page 7. All symbols have also been defined. |
Reviewer 2 Report
Comments and Suggestions for Authors
The study regarding large-amplitude ring gyro is very interesting and maybe helpful for electrode design and control. I have only one concern about the assumption of linear electrostatic tuning for this large-amplitude gyro. For a small-amplitude gyro, the linear assumption might be convinciable, however, the most interesting part of this study is about the dynamic behavior when gyro is driven with a large amplitude. Would the author be able to explain more about how this assumption is validated in the analysis?
Author Response
Comments |
Response |
For a small-amplitude gyro, the linear assumption might be convinciable, however, the most interesting part of this study is about the dynamic behavior when gyro is driven with a large amplitude. Would the author be able to explain more about how this assumption is validated in the analysis? |
In practice, the linear assumption is well-approximated using low amplitudes and the tuning conditions (17) and (18) are independent of drive amplitude. However, in principle some degree of nonlinearity is present for any finite amplitude. One the aims of this work is to apply this assumption and the associated tuning conditions to larger amplitude regimes to test the resilience of the 8 and 16 electrode arrangements to electrostatic nonlinearities, accepting that the linear assumption is not valid at larger amplitudes. Lines 258-261 have been added to discuss how the linear electrostatic tuning condition is achieved in practice under linear, low amplitude operations. A reference has also been included. |
Reviewer 3 Report
Comments and Suggestions for Authors
This manuscript presented the mathematical models to investigate effects of electrostatic nonlinearity on the rate sensing performance of imperfect ring-based CVG’s for devices having 8 and 16 evenly distributed electrodes. And the numerical analysis conducted with larger drive amplitude. The theoretical models were verified by FEM.
Comments:
- The content of the manuscript is too long. Some work from the manuscript has been presented at conferences[14], and it would be possible to shorten the article and focus on the extended and innovative parts.
- The paper emphasizes the imperfect ring. But different imperfect conditions weren’t presented in numerical analysis.
- The paper doesn’t provide the experiment results.
Author Response
Comments |
Response |
The content of the manuscript is too long. Some work from the manuscript has been presented at conferences[14], and it would be possible to shorten the article and focus on the extended and innovative parts.
|
The work presented in the conference paper [16] focuses only on sense force balancing applied to the 8-electrode arrangement. This part of the work is an essential part of the current manuscript because one of the main aims of this work is to make comparisons of the conditions and effectiveness of sense force balancing between the 8 and 16 electrode arrangements, so both sets of results are essential. The longer length of the manuscript is attributed to the 2-part analysis (linear electrostatic tuning and sense for balancing), but is well within the range for articles published by Sensors. It should be noted, however, that this revised version is slightly longer to address comments from the other reviewers. |
The paper emphasizes the imperfect ring. But different imperfect conditions weren’t presented in numerical analysis. |
The aim of the work is to provide a theoretical framework for investigating the impact of nonlinearity, rather than a thorough investigation of the various imperfection sources. The work considered one type of imperfection (elastic) to demonstrate the applicability of the approaches used and can be extended to other types. |
The paper doesn’t provide the experiment results.
|
As of now, we are unable to conduct physical experiments. However, the finite element method used is well suited to validating the theoretical results. |
Reviewer 4 Report
Comments and Suggestions for Authors
This paper investigates the imperfect ring-based Coriolis Vibrating Gyroscopes. The equation of motion is established taking into account the nonlinear electrostatic force. The influence of electrostatic force on the performance of gyroscope is analyzed through theoretical analysis and finite element verification. The article is written at a good level. I think it can be published after minor revisions. The comments are as follows.
On line 263, equation 22 is obtained using the averaging method. The application of the averaging method here seems to be significantly different from that of A.H.Nayfeh, so it is recommended to give the detailed process or references.
Finite element analysis is very important for this article. On line 366, the study of mesh convergence is mentioned, and I think the results should be given. What are the boundary conditions of the ring, especially the support conditions.
On line 507, Error! Reference source not found.
Author Response
Comments |
Response |
On line 263, equation 22 is obtained using the averaging method. The application of the averaging method here seems to be significantly different from that of A.H.Nayfeh, so it is recommended to give the detailed process or references.
|
The averaging method is first mentioned in the manuscript on lines 271-272 on page 9, after equation (19). References [20, 21] to the averaging method have been added. |
Finite element analysis is very important for this article. On line 366, the study of mesh convergence is mentioned, and I think the results should be given. What are the boundary conditions of the ring, especially the support conditions. |
Results of mesh convergence have not been included in the manuscript due to concerns over the length of the manuscript, which have been echoed by another reviewer in this round of review. However, the mesh convergence results are given in a thesis (see Appendix C in [1]). Point forces are used to model the supports to avoid the increased mesh elements associated with geometrically modelling the 8 supports. Lines 388 – 391 on page 13 have been added to clarify this. |
On line 507, Error! Reference source not found. |
This error has been removed. |
Round 2
Reviewer 3 Report
Comments and Suggestions for Authors
This manuscript investigates the effect of electrostatic nonlinearity on the rate sensing performance of imperfect ring-based Coriolis Vibrating Gyroscopes (CVG’s) for devices having 8 and 16 evenly distributed electrodes. Mathematical models are developed for CVG’s operating in open loop and closed loop, and the effects of electrostatic nonlinearity are investigated for increasingly large drive amplitudes. The numerical analysis conducted with larger drive amplitude, and the theoretical models were verified by FEM.
The Submission has been greatly improved and is worthy of publication.